# BCL-2 and BOK regulate apoptosis by interaction of their C-terminal transmembrane domains

Tobias B Beigl [1], Alexander Paul [1], Thomas P Fellmeth [2], Dang Nguyen [3,4], Lynn Barber [1], Sandra Weller [1], Benjamin Schäfer [1], Bernhard F Gillissen [5], Walter E Aulitzky [6], Hans-Georg Kopp [1,6], Markus Rehm [7,8], David W Andrews [3,4,9], Kristyna Pluhackova [2] & Frank Essmann [1,10]✉

## Abstract

The Bcl-2 family controls apoptosis by direct interactions of pro- and anti-apoptotic proteins. The principle mechanism is binding of the BH3 domain of pro-apoptotic proteins to the hydrophobic groove of anti-apoptotic siblings, which is therapeutically exploited by approved BH3-mimetic anti-cancer drugs. Evidence suggests that also the transmembrane domain (TMD) of Bcl-2 proteins can mediate Bcl-2 interactions. We developed a highly-specific split luciferase assay enabling the analysis of TMD interactions of pore-forming apoptosis effectors BAX, BAK, and BOK with anti-apoptotic Bcl-2 proteins in living cells. We confirm homotypic interaction of the BAX-TMD, but also newly identify interaction of the TMD of anti-apoptotic BCL-2 with the TMD of BOK, a peculiar pro-apoptotic Bcl-2 protein. BOK-TMD and BCL-2-TMD interact at the endoplasmic reticulum. Molecular dynamics simulations confirm dynamic BOK-TMD and BCL-2-TMD dimers and stable heterotetramers. Mutation of BCL-2-TMD at predicted key residues abolishes interaction with BOK-TMD. Also, inhibition of BOK-induced apoptosis by BCL-2 depends specifically on their TMDs. Thus, TMDs of Bcl-2 proteins are a relevant interaction interface for apoptosis regulation and provide a novel potential drug target.

**Keywords** BOK; BCL-2; Transmembrane Domain; Endoplasmic Reticulum; Apoptosis

**Subject Category** Autophagy & Cell Death

## Introduction

Proteins of the Bcl-2 family constitute the central hub for intracellular regulation of cell survival and demise by either preventing or promoting apoptotic cell death. Classically, the Bcl-2 protein family is divided into three subgroups according to their structure and function in apoptosis signaling (Fig. EV1A, reviewed in (Kale et al, 2018b)): Pro-survival/anti-apoptotic Bcl-2 family proteins (i) (including BCL-2, BCL-XL, BCL-W, MCL-1, and BFL-1/A1) interact with and inhibit pro-apoptotic siblings by accommodating their BH3 domain, one of the four different Bcl-2 homology (BH) domains in a hydrophobic groove of the anti-apoptotic counterparts which results in mutual sequestration. Pro-apoptotic multidomain effector proteins (ii) BAX, BAK, and BOK, once activated, oligomerize and initialize cell death by forming pores in the mitochondrial outer membrane (MOM). The third group, the so-called BH3-only proteins (iii), can be split into two subgroups according to their proposed mode of action: Sensitizer BH3-only proteins like BAD and NOXA act by occupying the hydrophobic groove of anti-apoptotic Bcl-2 proteins thereby blocking their capacity to hold active effector proteins in check. Activator BH3-only proteins like BID and BIM, however, allegedly directly interact with the pore-forming effectors and induce their conversion into an active conformation. Since active effectors can be blocked by anti-apoptotic Bcl-2 proteins, cells are ultimately sentenced to death once the anti-apoptotic capacity is exhausted and oligomerization of effector proteins mediates MOM permeabilization (MOMP). Upon MOMP, pro-apoptotic factors are released from mitochondria, importantly cytochrome c, which instigates activation of apoptosis-specific proteases (caspases) and destruction of the cell.

The BH3 domain and hydrophobic groove are recognized as important interaction sites of Bcl-2 proteins. Hence, the individual amino acid sequence of specific BH3 domains and amino acid composition of the hydrophobic groove in different Bcl-2 proteins cause variations in their mutual affinity and thus constitute the basis of the complex interaction network of Bcl-2 proteins (Banjara et al, 2020; Osterlund et al, 2022). Importantly, elucidation of the structural basis for Bcl-2 protein interaction inspired development of small-molecule drugs that mimic pro-apoptotic BH3 domains to specifically bind to the hydrophobic groove of anti-apoptotic Bcl-2 proteins and block their activity. These BH3 mimetics effectively exploit the Bcl-2 interaction network to push cancer cells into apoptosis.

[1]Robert Bosch Center for Tumor Diseases, Stuttgart, Germany. [2]Cluster of Excellence SimTech, University of Stuttgart, Stuttgart, Germany. [3]Department of Medical Biophysics, Faculty of Medicine, University of Toronto, Toronto, Canada. [4]Biological Sciences Platform, Sunnybrook Research Institute, Toronto, Canada. [5]Department of Hematology, Oncology, and Tumorimmunology, Charité University Medicine, Berlin, Germany. [6]Robert-Bosch-Hospital, Stuttgart, Germany. [7]Institute of Cell Biology and Immunology, University of Stuttgart, Stuttgart, Germany. [8]Stuttgart Research Center Systems Biology, University of Stuttgart, Stuttgart, Germany. [9]Department of Biochemistry, Faculty of Medicine, University of Toronto, Toronto, Canada. [10]Department of Molecular Medicine, Interfaculty Institute for Biochemistry, Eberhard Karls University Tübingen, Tübingen, Germany. ✉E-mail: frank.essmann@bosch-health-campus.com

Increasingly detailed elucidation of Bcl-2 protein interaction refined models of the Bcl-2 regulatory network from the initial "rheostat" to "direct activation" (Letai et al, 2002), "embedded together" (Leber et al, 2007), "unified" (Llambi et al, 2011), and "hierarchical" (Chen et al, 2015) model. In addition to mutual binding via BH3 domain and hydrophobic groove, the dynamic Bcl-2 interaction is modulated by protein abundance, post-translational modification, and subcellular localization (Kale et al, 2018b). The most important site where Bcl-2 proteins exert their function, i.e., regulation of cytochrome c release, are mitochondria. Consequently, composition of the MOM proposedly affects membrane interaction with Bcl-2 proteins and membrane insertion. Interestingly, the mitochondria-specific membrane component cardiolipin proposedly "glues" together BAX transmembrane domains during oligomerization (Lai et al, 2019). A C-terminal transmembrane domain (TMD) is present in most Bcl-2 proteins which was early recognized to target the family members to specific intracellular membranes (Zhu et al, 1996; Horie et al, 2002). Although generally referred to as TMD, the individual Bcl-2 C-termini insert into the lipid bilayer to varying extent. In general, TMDs consist of roughly 20 amino acids that form a single-pass α-helix flanked by charged amino acids on either side (Schinzel et al, 2004). In contrast, tail-anchor sequences tend to have fewer contiguous hydrophobic residues (~15) followed by less than 12 more hydrophilic residues (Brito et al, 2019). Naturally, amino acid sequence of the TMD crucially affects subcellular localization of Bcl-2 proteins. For example, specific targeting of BAK or BCL-XL to MOM depends on flanking basic amino acid side chains as well as on helix hydrophobicity (Kaufmann et al, 2003). TMDs were shown to be a critical structural feature in Bcl-2 family proteins in several studies (Jeong et al, 2004; Guedes et al, 2013; Stehle et al, 2018; Chi et al, 2020) since absence, mutation, or post-translational modification of TMDs substantially affects targeting and function of Bcl-2 proteins (Nechushtan et al, 1999; Gardai et al, 2004; Simonyan et al, 2016; Kale et al, 2018a; Lucendo et al, 2020).

Increasing evidence indicates that direct TMD-TMD interaction of Bcl-2 proteins critically impacts on apoptosis regulation, for example, that enlargement of BAX-oligomeric pores depends on BAX dimer-dimer interaction via their TMDs (Zhang et al, 2016; Liao et al, 2016). Also, interaction of pro- and anti-apoptotic Bcl-2 proteins such as MCL-1 and BOK via TMDs modulates cell death regulation (Andreu-Fernández et al, 2017; Lucendo et al, 2020) stressing fine-tuning of the BH3 domain:hydrophobic groove-based interaction scheme. The BH3-only proteins BIML and PUMA bind to their anti-apoptotic counterparts BCL-2 and BCL-XL by both BH3 domain and TMD which substantially influences binding affinity and is described as a crucial "double-bolt lock" mechanism (Liu et al, 2019; Pemberton et al, 2023).

The first identified member and namesake of the family, the anti-apoptotic protein BCL-2, efficiently inhibits BAX and BAK pore formation in the MOM. However, in non-apoptotic cells a large fraction of BCL-2 resides at the ER, which mainly depends on the BCL-2 TMD (Kaufmann et al, 2003). Recently, TOM20-mediated translocation of BCL-2 from ER to mitochondria-associated membranes (MAMs) and mitochondria upon apoptosis induction was postulated by Lalier et al (Lalier et al, 2021). Andreu-Fernández et al showed, that BCL-2 TMD peptides not only form stable homodimers but also interact with BAX-TMD in biological membranes (Andreu-Fernández et al, 2017). Survival promotion by

BCL-2 takes place also at the ER since BCL-2 regulates calcium signaling by binding to inositol 1,4,5-trisphosphate receptors (IP3Rs) in ER membranes (Rong et al, 2009; Monaco et al, 2012; Chang et al, 2014).

In contrast to BAX and BAK, the pro-apoptotic protein BOK is localized at the ER (Echeverry et al, 2013). Also BOK possesses a C-terminal membrane-targeting sequence, termed as TMD, although no empirical data confirmed its membrane-spanning character. Early after identification of BOK a yeast two-hybrid assay suggested interaction of BOK with BFL1/A1 and MCL-1 (Hsu et al, 1997). The interaction site of BOK with anti-apoptotic MCL-1 was recently pinpointed to their TMDs which is to date the only described direct interaction of BOK with anti-apoptotic Bcl-2 proteins (Stehle et al, 2018; Lucendo et al, 2020). However, BOK-induced apoptosis was formerly shown not to be counteracted by any anti-apoptotic Bcl-2 family protein (Llambi et al, 2016). Thus, despite sporadic progress in illuminating Bcl-2 TMD interaction, the bulk of the TMD network and its fine tuning of apoptosis signaling remains in the dark, overshadowed by the dominant BH3 domain:hydrophobic groove interaction.

Interaction of Bcl-2 TMDs was studied by various molecular biological methods (Zhang et al, 2016; Andreu-Fernández et al, 2017; Lucendo et al, 2020). Also in silico modeling (Wassenaar et al, 2015) was recently used to study the homomeric and heteromeric interactions in transmembrane dimers of MCL-1, BAK, and BAX in an artificial phosphocholine membrane (Lucendo et al, 2020).

Here, employing a novel split luciferase assay, we reveal direct interaction of BCL-2-TMD with TMD of pro-apoptotic effector protein BOK. Fluorophore-conjugated TMD peptides and full-length proteins co-localize at the ER. This TMD-directed co-localization is independent of the BH3 domain. High-throughput multiscale molecular dynamics (MD) simulations of BCL-2-TMD/BOK-TMD oligomerization in an ER membrane mimic support interaction of BOK-TMD and BCL-2-TMD. Simulations propose dynamic heterodimers and -trimers, but unexpectedly also stable tetramers of two BOK-TMDs and two BCL-2-TMDs, suggesting that BOK is retained in the ER membrane by BCL-2. Based on MD simulations we identified L223, V226, and I230 in BCL-2 as key residues for BOK-TMD/BCL-2-TMD interaction. Functionally, BCL-2 inhibits BOK-induced apoptosis and consequently TMD amino acid sequence in BOK and BCL-2 critically affect inhibition by BCL-2. The newly identified interaction of the TMDs of BOK and BCL-2 is a new component of the regulatory interaction network of Bcl-2 family proteins representing BH3 domain-independent regulation of apoptosis.

# Results

## BCL-2-TMD and BOK-TMD interaction revealed by a novel bimolecular split luciferase assay

Several studies proposed interaction of selected Bcl-2 proteins via their TMDs such as BAX and BCL-XL (Todt et al, 2013; Andreu-Fernández et al, 2017; Lucendo et al, 2020). Since these reports show a mere fraction of TMD interaction among Bcl-2 proteins, we initially set out to systematically analyze interaction of the TMDs in anti-apoptotic and pro-apoptotic effector Bcl-2 proteins. We

designed a bimolecular split luciferase assay based on the NanoBiT system which has the advantage of (a) very low background signal because the subunits of NanoBiT, large BiT (LgBiT) and small BiT (SmBiT), have low affinity and (b) its applicability in living cells (England et al, 2016). The generated plasmids encode for LgBiT or SmBiT—fused by a hydrophilic linker sequence to the TMD of interest (Fig. 1A,B). Upon expression, interaction of a respective pair of TMDs brings LgBiT and SmBiT in close proximity and allows for formation of the functional luciferase which then can process the substrate coelenterazine-h to produce a luminescence signal. Additionally, plasmids encode for preceding fluorophores mCitrine (LgBiT-TMD) or mTurquoise2 (SmBiT-TMD), respectively, separated by a T2A self-cleaving sequence from the NanoBiT-TMD to allow normalization of the luminescence signal to the fluorescence of mCitrine and/or mTurquoise2 (Fig. 1B).

To verify functionality of the split luciferase assay, we transfected HEK293FT cells with combinations of LgBiT-BAX-TMD and either SmBiT-BAX-TMD or SmBiT-TOM5-TMD and analyzed luminescence, fluorescence and protein expression. Since BAX pores enlarge by BAX-TMD dimerization (Zhang et al, 2016; Liao et al, 2016), the BAX-TMD homotypic interaction served as a positive control. The TMD of TOM5, a component of the translocase of the outer membrane (TOM) mitochondrial import complex, served as a negative control non-binding partner for the BAX-TMD. Indeed, luminescence detected from co-expression of SmBiT-TOM5-TMD with LgBiT-BAX-TMD was similar to the background signal (background 185.9 RLU vs. 303.6 RLU). The co-expression of SmBiT-BAX-TMD and LgBiT-BAX-TMD resulted in a strong luminescence signal (16572.0 RLU) in line with homotypic interaction (Fig. 1C, upper panel). Simultaneous expression of fluorescent proteins was confirmed by confocal laser-scanning microscopy (cLSM) and flow cytometry (Figs. 1D and EV1B) showing a proportional signal of mCitrine and mTurquoise2 fluorescence in cells co-transfected with equal amounts of LgBiT-BAX-TMD (mCitrine) and SmBiT-BAX-TMD or SmBiT-TOM5-TMD (mTurquoise2). In an additional control experiment, titration of SmBiT:LgBIT ratio resulted in a peak luminescence signal at a ratio of 1:1 suggesting efficient homotypic dimerization of BAX-TMDs (Fig. EV1C). Moreover, Western blot confirmed expression of LgBiT-fused TMDs and expression of mTurquoise2 as a surrogate marker for expression of SmBiT-TMD (Fig. 1C, lower panel). Next, we simultaneously analyzed fluorescence and luminescence in a multimode plate reader. Specific mCitrine or mTurquoise2 fluorescence was detected in HEK293 cells transfected with respective plasmids (Fig. 1E). The signal for mCitrine and mTurquoise2 was comparable in 1:1 co-transfected cells (Fig. 1E) and also normalization of luminescence via either fluorophore is comparable (Fig. EV1D). Therefore, in subsequent experiments luminescence is normalized to fluorescence of mTurquoise2.

After successful validation of the split luciferase system, we next analyzed interaction of TMDs from Bcl-2 effector proteins, BAX, BAK and BOK, with TMDs of anti-apoptotic proteins BCL-2, BCL-XL, and MCL-1 in HEK293FT cells. In these co-transfection experiments, we set the normalized luminescence of BAX-TMD homotypic interaction to 1. Combining LgBiT-TMD of effectors with SmBiT-TMD of anti-apoptotic Bcl-2 proteins we found relative normalized luciferase activity for combination of BCL-2-TMD with BAX-TMD (0.03) or BAK-TMD (0.16) (Fig. 1F) which

was comparable to the negative control (BAX-/TOM5-TMD, 0.05). Significant luminescence was detected for the combination of BCL-2-TMD with BOK-TMD reaching 46% of the positive control (BAX-/BAX-TMD) luminescence (Fig. 1F). The interaction of BOK-TMD with BCL 2-TMD was confirmed in MCF-7 cells with a relative normalized luminescence of 30% compared to the positive control (Fig. EV1E). In addition, all effector TMDs produced a luminescence signal when co-expressed with BCL-XL-TMD indicating promiscuous interaction (Fig. 1F).

Taken together, we developed and validated a plasmid-based, bimolecular split luciferase assay to analyze interaction of TMDs from Bcl-2 family proteins and revealed a novel direct interaction of the BCL-2-TMD with BOK-TMD.

## BCL-2-TMD and BOK-TMD co-localize at the ER

As co-localization is a prerequisite for interaction, we next investigated the subcellular localization of TMDs fused to fluorescent proteins. We assumed that the exposed TMD sequence in Bcl-2 proteins is sufficient to target the protein to a specific subcellular membrane as shown for ER-like localization of BCL-2-TMD fused with enhanced green fluorescent protein (EGFP) (Egan et al, 1999; Kaufmann et al, 2003). Co-localization of fluorophore-coupled TMD peptides with fluorescent organelle markers then allows estimation of TMD (and protein) subcellular localization.

MCF-7 cells that express subcellular markers for ER or mitochondria (see Methods) were transfected with plasmids for the expression of mTurquoise2 with C-terminal fused TMD. The subcellular localization of the TMD peptides was then imaged by cLSM. BAX-TMD and BAK-TMD were predominantly localized to the mitochondria, while BOK-TMD and BCL-2-TMD both co-localized with the ER marker (Fig. 2A,B). BCL-XL-TMD peptides co-localized with the mitochondria, although some overlap with the ER was observed (Fig. EV2A,B). Surprisingly, MCL-1-TMD was primarily localized to the ER, but also in the cytosol (Fig. EV2A,B). We quantified the co-localization by calculating the Pearson's correlation coefficient for individual cells between TMD peptides and ER- and mitochondrial markers, respectively (Figs. 2C and EV2C,D). BOK-TMD and BCL-2-TMD showed the highest correlation coefficient with the ER among the TMDs tested with a mean of $r_{BOK-TMD/ER} = 0.50$ and $r_{BCL-2-TMD/ER} = 0.42$ comparable to the ER-specific cb5-TMD peptide (mean $r_{cb5-TMD/ER} = 0.40$).

To further validate the subcellular localization of BAX-TMD, BAK-TMD, and BOK-TMD, we generated cell lines from $BAX^{-/-}/BAK^{-/-}$ baby mouse kidney cells (BMK/DKO) (Degenhardt et al, 2002) exogenously expressing mTurquoise2-conjugated N-terminally to BAX-TMD, BAK-TMD, or BOK-TMD. In BMK/DKO cells, BAX-TMD and BAK-TMD displayed a mitochondria-like distribution, whereas BOK-TMD showed an ER-like distribution as imaged by confocal microscopy (Fig. 2D). For high-content analysis of TMD co-localization with mitochondria and ER markers, BMK/DKO cell lines were stained with Draq5 (nuclei), Mitotracker red (mitochondria), and BODIPY-Thapsigargin (ER) before imaging., the Pearson's correlation coefficient (r) in the cytoplasmic region (excluding nuclei) of at least 1000 individual cells was calculated (Fig. 2E). As a positive control for ER localization, a BMK/DKO cell line stably expressing mCerulean3-BIK, an ER-localized BH3-only protein (Osterlund et al, 2023), was

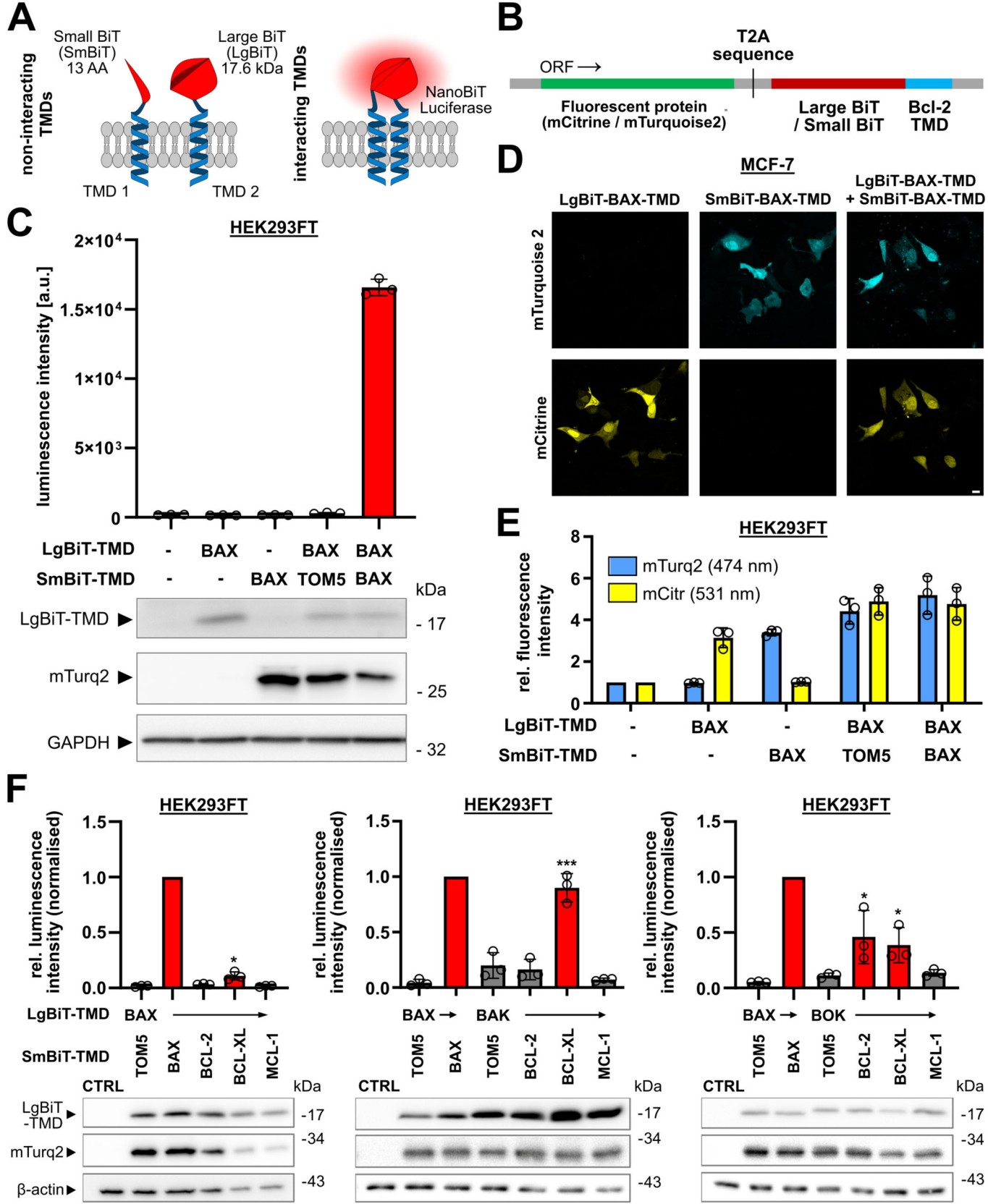

◀ **Figure 1. A bimolecular split luciferase assay reveals that TMDs of BCL-2 interact with TMDs of BOK but not with TMDs of BAX or BAK.**

(A) Schematic of developed split luciferase assay and its working principle. (B) Schematic of plasmid insert structure used for NanoBiT-TMD fusion protein expression featuring simultaneous expression of mCitrine (LgBiT-TMDs) or mTurquoise2 (SmBiT-TMDs). (C) Absolute luminescence intensity of HEK293FT cells transfected with plasmids for the expression of LgBiT- and/or SmBiT-TMDs (or left untransfected, upper panel) in the split luciferase assay. Cells were harvested after 24 h and luminescence was detected in a multimode plate reader. Mean ± sd of technical triplicates from one representative experiment. Red bar color indicates significant difference to BAX-TMD/TOM-5-TMD negative control. Whole cell lysates of same cells were analyzed for expression of LgBiT and mTurquoise2 by Western blot (lower panel). Representative blot from two independent experiments. GAPDH was used as loading control. (D) Specific expression of mCitrine and mTurquoise2 from NanoBiT-TMD plasmids. MCF-7 cells were transiently transfected with plasmids for the expression of LgBiT-BAX-TMD, SmBiT-BAX-TMD or a combination of both. Cells were fixed after 18 h followed by cLSM. Images validate specific detection of mTurquoise2 and mCitrine. Representative images out of three independent experiments. Scale bar = 10 μm. (E) Specific fluorescence of mCitrine and mTurquoise2 in the split luciferase assay. mCitrine and mTurquoise2 fluorescence intensity of cells transfected as in E detected by a multimode plate reader. Intensities are shown relative to untransfected control as mean ± sd of three independent experiments. (F) TMDs of all effector proteins interact with BCL-XL-TMD, while only BOK-TMD interacts with BCL-2-TMD. Split luciferase assay in HEK293FT cells transfected with plasmids for the expression of indicated NanoBiT-TMDs. Cells were harvested after 24 h and samples were both used for split luciferase assay and Western blot. BAX-TMD/TOM5-TMD serves as a negative control, while BAX-TMD/BAX-TMD serves as positive control. SmBiT-TMDs of anti-apoptotic Bcl-2 proteins (BCL-2/BCL-XL/MCL-1) were combined with LgBiT-BAX-TMD (left), LgBiT-BAK-TMD (middle), or LgBiT-BOK-TMD (right). Graphs show luminescence intensity relative to the positive control and normalized to mTurquoise2 fluorescence intensity. Shown is the mean ± sd from three independent experiments. In blots below, corresponding whole cell lysates were analyzed for expression of LgBiT and mTurquoise2 (mTurq2). Representative blots from three independent experiments are shown. Red bar color indicates significant difference to BAX-TMD/TOM-5-TMD negative control. *$p < 0.05$, ***$p < 0.001$, one-way ANOVA with Tukey's multiple comparison test. Source data are available online for this figure.

included. Expectedly, BAX-TMD and BAK-TMD showed predominant localization at the mitochondria (mean $r_{BAX\text{-}TMD/Mito} = 0.81$ and $r_{BAK\text{-}TMD/Mito} = 0.79$), whereas mitochondrial localization of BIK and BOK-TMD was less pronounced (mean $r_{BOK\text{-}TMD/Mito} = 0.55$, $r_{BIK/Mito} = 0.46$). On the other hand, BAX-TMD and BAK-TMD displayed a less significant localization at ER (mean $r_{BAX\text{-}TMD/ER} = 0.64$ and $r_{BAK\text{-}TMD/ER} = 0.76$) while in contrast, BIK and BOK-TMD strongly localized to the ER (mean $r_{BOK\text{-}TMD/ER} = 0.92$ and $r_{BIK/ER} = 0.88$). Compartment-specificity was validated by comparison of BODIPY-Thapsigargin and Mitrotracker green to Mitotracker red (Fig. EV2E). Thus, in BMK/DKO cells, BAX-TMD and BAK-TMD preferentially localize at the mitochondria, whereas BOK-TMD associates with the ER.

Since we found interaction of BOK-TMD and BCL-2-TMD in the split luciferase assay, we next analyzed co-localization of mCitrine-BOK-TMD and mTurquoise2-BCL-2-TMD in MCF-7 cells. Using Mitotracker staining or co-expression of mCarmine fused to the ER-targeting sequence of Calreticulin as an ER marker (Fabritius et al, 2018), we found visible and quantifiable co-localization of BOK-TMD and BCL-2-TMD at the ER with $r_{BOK\text{-}TMD/BCL\text{-}2\text{-}TMD} = 0.52$ (Fig. 3A,B).

These analyses show that BAX-TMD and BAK-TMD both localize to the mitochondria providing a valid explanation for their poor interaction with ER-localized BCL-2-TMD. Furthermore, BOK-TMD and BCL-2-TMD co-localize at the ER giving a rational basis for the newly identified interaction.

## TMDs of BOK and BCL-2 are critical for their co-localization at the ER

Classically, interaction of full-length BCL-2 with BAX and BAK is understood to result from binding of the BH3 domain of BAX and BAK to the hydrophobic groove of BCL-2 (Ding et al, 2010; Willis et al, 2005). However, TMD-mediated interaction of Bcl-2 family proteins has been reported repeatedly (Todt et al, 2013; Andreu-Fernández et al, 2017; Lucendo et al, 2020). In order to tackle the question to which extent the interaction between BH3 domain and hydrophobic groove impacts co-localization of BCL-2 and effector proteins BAX, BAK, and BOK, we utilized vectors for the expression of EGFP-fused full-length proteins BAX, BAK and

BOK and chimeric BCL-2 proteins with swapped transmembrane domains (schematic in Fig. 4A,B).

MCF-7 cells with labeled mitochondria or ER that expressed EGFP-tagged full-length BAX, BAK or BOK were imaged by cLSM. Image analysis of cells which did not show clustered EGFP-signals revealed cytosolic and partially mitochondrial localization of BAX, exclusively mitochondrial localization of BAK, and ER localization of BOK (Figs. 4C and EV3A,C). Interestingly, mCherry-BCL-2 showed co-localization with each EGFP-fused effector protein (Figs. 4D and EV3B). However, co-localization of EGFP-BAX and EGFP-BAK with mCherry-BCL-2 was detected at the mitochondria ($r_{BAX/BCL\text{-}2} = 0.34$, $r_{BAK/BCL\text{-}2} = 0.52$), whereas EGFP-BOK and mCherry-BCL-2 co-localized at the ER ($r_{BOK/BCL\text{-}2} = 0.47$). To challenge the role of the BH3 domain:hydrophobic groove interaction as mediator of co-localization, we analyzed co-localization of chimeric mCherry-BCL-2 harboring the TMD sequence of cytochrome b5 (cb5) or TOM5 with EGFP-fused effectors BAX, BAK, and BOK. Exchange of the TMD in BCL-2 to the ER-targeting cb5-TMD reduced co-localization of BCL-2 with BAX and BAK, respectively, yet only in cells expressing low levels of BAX or BAK, respectively, while co-localization with BOK remained unchanged ($r_{BOK/BCL\text{-}2\text{-}cb5\text{-}TMD} = 0.44$, Figs. 4E and EV3D,F). Interestingly, in cells with BAX/BAK-clustering at the mitochondria, BCL-2$^{cb5\text{-}TMD}$ co-localized with these clusters ($r_{BAX/BCL\text{-}2\text{-}cb5\text{-}TMD} = 0.44$, $r_{BAK/BCL\text{-}2\text{-}cb5\text{-}TMD} = 0.54$, Figs. 4E and EV3D). Thus, we assume that interaction of the BCL-2 hydrophobic groove with the BH3 domain of active BAX and BAK is dominant over BCL-2-TMD-mediated localization and active BAX and BAK attract BCL-2$^{cb5\text{-}TMD}$ to mitochondria. As expected, mCherry-BCL-2 with a conjugated TOM5-TMD was strictly localized to the mitochondria, effectively disrupting the co-localization with BOK ($r_{BOK/BCL\text{-}2\text{-}TOM5\text{-}TMD} = -0.08$, Figs. 4F and EV3E). Hence, while the co-localization of BCL-2 with BAX or BAK clusters at the mitochondria is BH3 domain-driven, BCL-2 localization at the ER and co-localization with BOK depends on the TMD rather than on the BH3 domain.

Corroborating this conclusion, mutation of the conserved leucine of the BH3 domain in EGFP-BOK (L70E mutation) did not abrogate co-localization with mCherry-BCL-2 ($r_{BOK\ L70E/BCL\text{-}2} = 0.75$, Figs. 4G and EV3G). In contrast, deletion of the BOK-TMD from mCherry-BOK (mCherry-BOK$^{\Delta TMD}$) resulted in a diffuse cytosolic localization of BOK indicating inability of BOK$^{\Delta TMD}$ to integrate into membranes

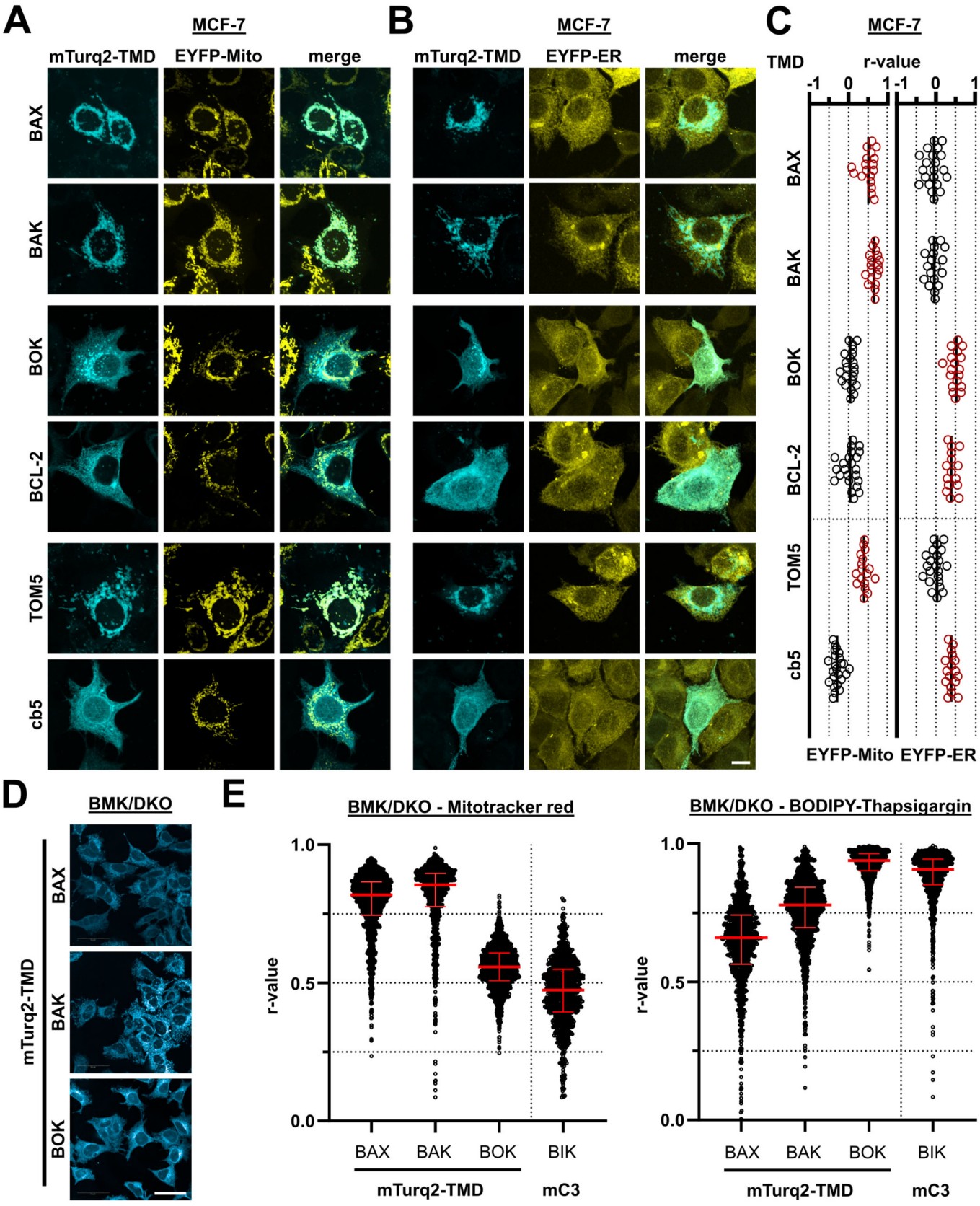

**Figure 2. BCL-2-TMD and BOK-TMD are predominantly ER-localized.**

(A, B) Subcellular localization of TMD peptides. MCF-7 cells stably expressing (A) EYFP-Mito or (B) EYFP-ER were transfected with plasmids for the expression of mTurquoise2 (mTurq2)-labeled Bcl-2 TMD peptides. Cells were fixed after 24 h and TMD localization was analyzed by cLSM. Images are maximum projection of z-stacks representative of three independent experiments. Scale bar = 10 μm. (C) Quantitative analysis of the co-localization from cLSM in (A, B) (three independent experiments). Scatter plots show Pearson's r correlation coefficients of in total ≥15 single cells. Mean is marked as a vertical line. Data with higher Pearson's r (for EYFP-Mito or EYFP-ER) is colored red. (D) Representative spinning disk confocal images of BMK/DKO cells expressing mTurquoise2-fused BAX-TMD, BAK-TMD, or BOK-TMD. Scale bar = 50 μm. (E) Subcellular localization analysis of effector TMD peptides in BMK/DKO cells. BMK/DKO cell lines expressing mTurquoise2-BAX-TMD, -BAK-TMD, and -BOK-TMD were labeled with DRAQ5 (nuclei) and Mitotracker red (left) or BODIPY-Thapsigargin (right). Images were acquired using an Opera Phenix spinning disk microscope and Pearson's r was calculated. BAX-TMD and BAK-TMD predominantly localize to mitochondria, while BOK-TMD localizes to ER. Shown is median + IQR of n ≥ 1000 cells. Source data are available online for this figure.

(Fig. 4H). Co-expression of mCherry-BOK$^{\Delta TMD}$ with EGFP-BCL-2 did not change the diffuse localization of mCherry-BOK$^{\Delta TMD}$ (Figs. 4H and EV3H). Thus, also TMD removal from BOK effectively abolishes co-localization with BCL-2 ($r_{BOK-\Delta TMD/BCL-2} = -0.17$).

We conclude that co-localization of BCL-2 with BAX and BAK at mitochondria depends on exposure of the active effector's BH3 domain. In contrast, analogous to BOK-TMD and BCL-2-TMD, also full-length BOK and BCL-2 co-localize at the ER, which does not depend on the BH3 domain of BOK but is driven by the TMD of BOK and BCL-2.

## Molecular dynamics reveals BOK-TMD and BCL-2-TMD interactions in the ER membrane

Since the identified co-localization of BCL-2 and BOK at the ER corroborated the TMD interaction found in the split luciferase assay, we set out to investigate the interaction of BOK-TMD with BCL-2-TMD at molecular resolution by high-throughput MD simulations (Wassenaar et al, 2015). Because lipid composition of biomembranes substantially modulates interaction of transmembrane proteins (Pluhackova et al, 2016) we prepared a mimic (Pluhackova and Horner, 2021) of ER membrane (described in the Appendix) and studied homo- and hetero-oligomerization of BCL-2-TMD and BOK-TMD.

### BOK-TMD/BOK-TMD homodimerization and BOK-TMD/BCL-2-TMD heterodimerization

MD simulations of spontaneous BOK-TMD homodimerization at coarse-grained (CG) resolution resulted in three main clusters (BOK-TMD/BOK-TMD-I, -II, and –III) of right-handed dimer structures (Fig. 5A; Appendix Figs. S1–S4, Appendix Table S9). In the most populated BOK-TMD/BOK-TMD-I comprising 55% of all CG structures, the two TMDs are shifted by ~1.5 nm along the helix axis and residues K202$^{BOK}$, A203$^{BOK}$, F206$^{BOK}$, V207$^{BOK}$, and P210$^{BOK}$ of one BOK-TMD interact with residues H188$^{BOK}$, V191$^{BOK}$, A192$^{BOK}$, C195$^{BOK}$, and R199$^{BOK}$ of the other BOK-TMD (Fig. 5A). The proximity of A192$^{BOK}$ from one and A203$^{BOK}$ from the other BOK-TMD enables tight dimer packing. This asymmetric interaction interface appeared stable over the course of atomistic simulations, indicating its specificity. The strong tilt of the peptides in the bilayer and their shift relative to each other enable the charged residues K202$^{BOK}$, R199$^{BOK}$, of individual BOK-TMD peptides to snorkel to the lipid headgroups of different membrane leaflets (Rabe et al, 2016; Korn and Pluhackova, 2022) (Appendix Fig. S5).

CG simulations of spontaneous BOK-TMD/BCL-2-TMD dimerization resulted in four clusters of heterodimers (Appendix

Figs. S1, S5–8, Appendix Table S10). The right-handed BOK-TMD/BCL-2-TMD-I, shown in Fig. 5B, was formed in 44%, the right-handed BOK-TMD/-BCL-2-TMD-II in 34%, the left-handed BOK-TMD/BCL-2-TMD-III in 8% and the right-handed BOK-TMD/BCL-2-TMD-IV in 7% of all dimers. With the exception of the structures from the rarely-formed BOK-TMD/BCL-2-TMD-IV cluster, all-atom simulations of BOK-TMD/BCL-2-TMD heterodimers have shown even larger conformational flexibility than atomistic simulations of BOK-TMD/BOK-TMD homodimers, resulting in significant reorientation of the two TMDs and even dissociation in 2 out of 12 simulations. Certain commonalities were identified for the crossed dimers BOK-TMD/BCL-2-TMD-I, II and III with the exception of the (nearly) dissociated conformations: BOK-TMD most often contacted L223$^{BCL-2}$, V226$^{BCL-2}$, or I230$^{BCL-2}$, the backbone of BCL-2-TMD typically crossed BOK-TMD at A203$^{BOK}$, and R199$^{BOK}$, K202$^{BOK}$, as well as K218$^{BCL-2}$ snorkeled to the cytosolic membrane leaflet causing its deformation and local indentation.

The number of interaction sweet spots in BOK-TMD/BOK-TMD and BOK-TMD/BCL-2-TMD dimers (Fig. 5C), which is an indicator of higher order oligomers (Han et al, 2016), taken together with high occurrence yet low conformational stability of BOK-TMD homodimers and even more strongly BOK-TMD/BCL-2-TMD heterodimers suggest, that BOK-TMD/BCL-2-TMD likely prefer to form oligomeric structures. To test this hypothesis, we have inserted two BOK-TMDs and two BCL-2-TMDs into the ER membrane mimic and studied their spontaneous association 50 times at CG level. After 50 μs 50% of the TMDs formed tetramers, 36% were trimers (in 13 simulations one BOK-TMD was stabilized by two BCL-2-TMD, and in 5 simulations two BOK-TMDs associated with one BCL-2-TMD), in 5% a heterodimer and in 2% BCL-2-TMD homodimer were formed. The trimers and tetramers were manually grouped in PyMOL (Schrödinger, 2023) and their stability studied by 1 μs AA simulation, each.

### 2x BOK-TMD/BCL-2-TMD and BOK-TMD/2x BCL-2-TMD trimers

Even though the most 2x BOK-TMD/BCL-2-TMD and BOK-TMD/2x BCL-2-TMD heterotrimers adopted a compact shape (Fig. 5D, Appendix Fig. S9) the heterotrimers appeared similarly dynamic as BOK-TMD /BCL-2-TMD heterodimers, as measured by their RMSD over AA simulations (Fig. 5E). The analysis of protein surface buried at the protein-protein interaction interfaces (Fig. 5G) has shown that in case of trimers 1.5 more protein surface is buried as compared to the dimer interfaces. Combined with our observation of compact trimers, where each peptide contacts two other peptides, this hints to the fact that upon trimerization smaller interaction surfaces between pairs of TMDs are formed. This can be

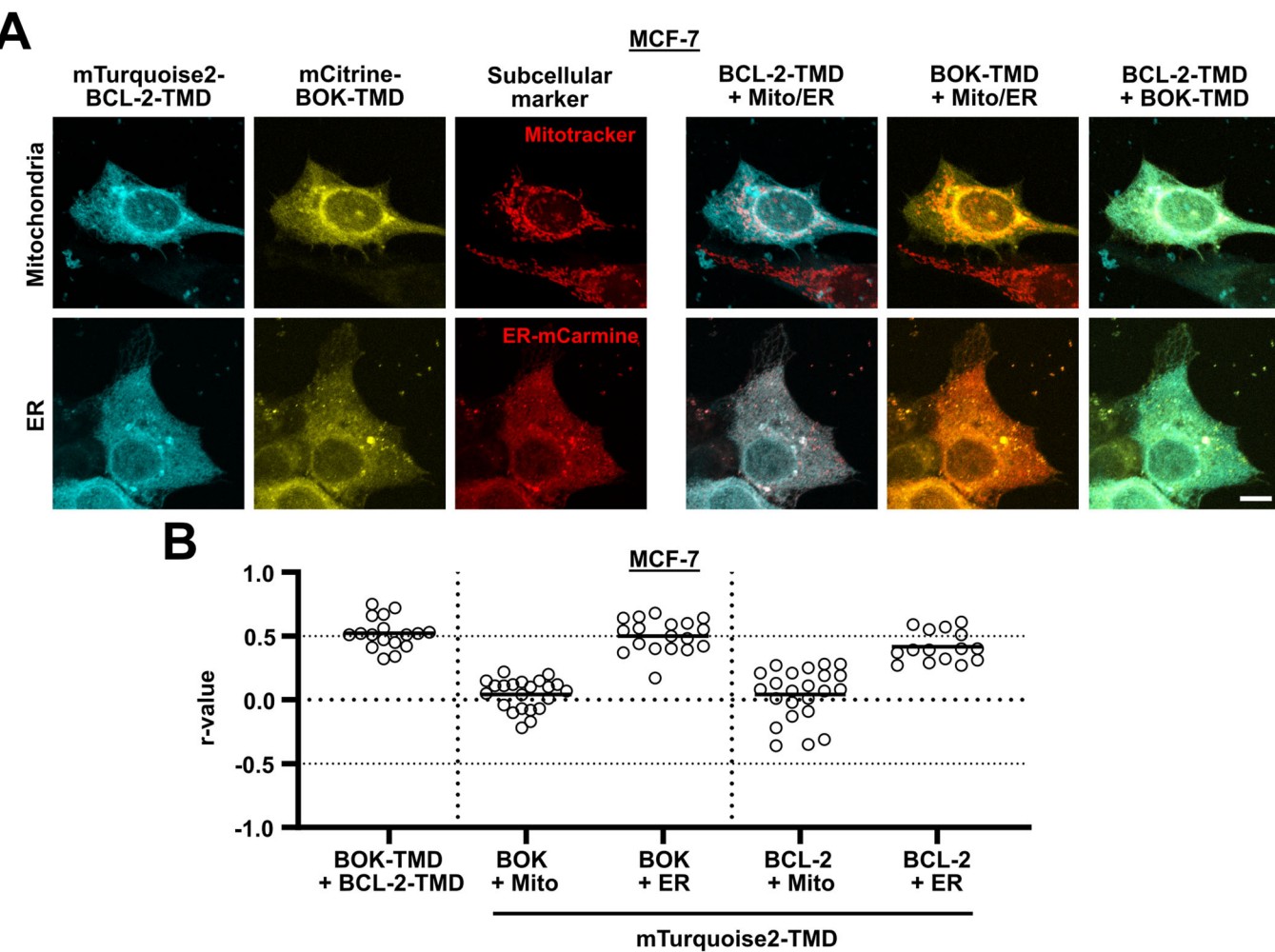

**Figure 3. BOK-TMD and BCL-2-TMD co-localize at the ER.**

(A) Co-localization of BOK-TMD and BCL-2-TMD at the ER. MCF-7 cells were transfected with plasmids for the expression of mTurquoise2-BCL-2-TMD and mCitrine-BOK-TMD. Cells were stained with Mitotracker red (upper panels) or co-transfected with plasmids for the expression of mCarmine-ER (lower panel). Cells were fixed after 24 h and TMD localization was analyzed by cLSM. Images are maximum projection of z-stacks representative of three independent experiments. Scale bar = 10 μm. (B) BCL-2-TMD and mCitrine-BOK-TMD co-localize at the ER. Quantitative analysis of co-localization between mTurquoise2-BCL-2-TMD and mCitrine-BOK-TMD from cLSM in (A). Scatter plots show Pearson's r correlation coefficients of in total ≥15 single cells from three independent experiments determined using Fiji software. Co-localization data between mTurquoise2-BCL-2-TMD/-BOK-TMD and EYFP-Mito and EYFP-ER from Fig. 2 is shown for comparison. Mean is marked as a horizontal line. Source data are available online for this figure.

explained by our visual observation of more crossed peptides in the trimers than in dimers where they typically form comparably long interaction surfaces.

### 2x BOK-TMD/2x BCL-2-TMD tetramers

Our CG simulations have shown tetramerization to be the preferred oligomerization state of two BOK-TMDs with two BCL-2-TMDs. Interestingly, BOK-TMDs and BCL-2-TMDs self-assembled into a great variety of heterotetramer conformations (Appendix Fig. S10). In 27% of the tetramers, the TMDs formed compact right-handed heterotetramers with slightly twisted shape enabling formation of extensive interaction interfaces. Three examples are shown in Fig. 5F. In 19% and 5% crossed dimers of parallel hetero- or homodimers, respectively, were formed, reminding of a hash shape. In 22% and 7% of the tetramers a dimer of parallel and crossed-

dimers arose (called halfhash from now on), which could be an intermediate between the former two tetramers (hash and twist). In 19% of the tetramers, one peptide was attached in a left-handed manner to a right-handed trimer. Atomistic simulations of the twist, hash, and half-hash tetramers have shown that the compact twist shape is the most favored (2/3 of the 18 AA simulations adopted this shape). Interestingly, in 4 other simulations hash or half-hash of homodimers appeared to be stable, contrary to the fact that all heterohashes and half-heterohashes rapidly evolved into compact twisted heterotetramers (Appendix Fig. S10). The increase of the hidden surface area in tetramers by factor 2 relative to the dimers (Fig. 5G) hints to further stabilization of the oligomer which is also confirmed by smallest RMSD values compared to dimers and trimers (Fig. 5E). In reality the stability of the tetramers compared to dimers and trimers is even higher than the RMSD plot suggest,

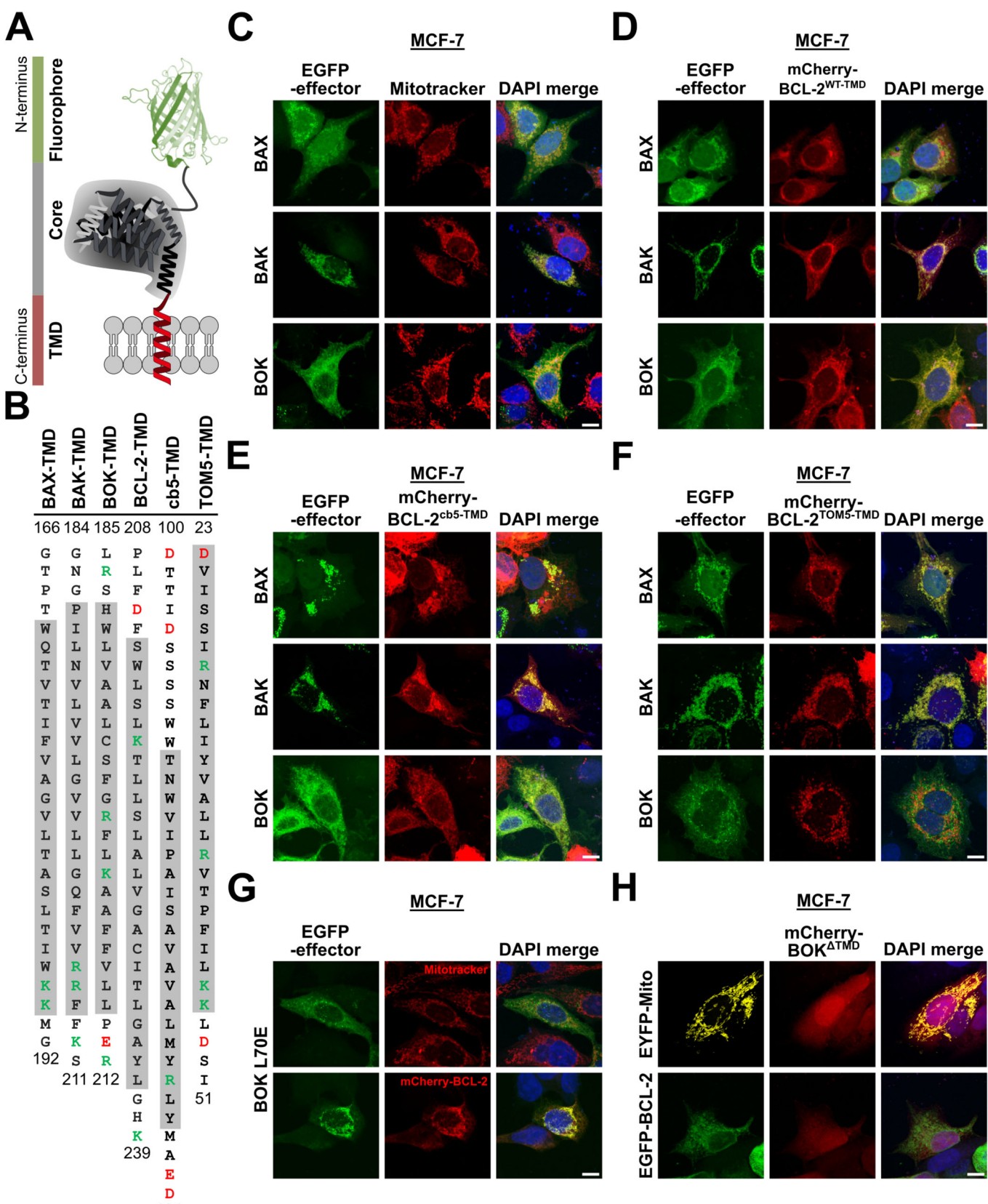

**Figure 4.  BOK and BCL-2 co-localization at the ER is dictated by their TMDs.**

(A) Schematic depiction of fluorophore-tagged full-length Bcl-2 proteins. (B) Amino acid sequences of BAX-, BAK-, BOK-, BCL-2-, cb5-, and TOM5-TMD. Positively charged amino acids are labeled green, negatively charged amino acids are labeled red. Alpha-helical sequence sections are marked in gray. (C–H) Co-localization of BAX, BAK, and BOK with (chimeric) BCL-2 protein showing that co-localization of BOK and BCL-2 is TMD-dependent. MCF-7 cells were transfected with plasmids for the expression of EGFP-BAX, EGFP-BAK, or EGFP-BOK. Cells were either stained with Mitotracker red (C) or co-transfected with plasmids for the expression of mCherry-BCL-2 wild-type (D, WT) or chimeric mCherry-BCL-2 proteins containing cb5-TMD (E) or TOM5-TMD (F). After 18 h, cells were fixed and mounted using DAPI-containing mounting medium followed by cLSM. (G) BOK L70E mutation does not alter BOK and BCL-2 co-localization. MCF-7 cells were transfected with plasmids for the expression of EGFP-BOK L70E. Cells were stained with Mitotracker red (upper panels) or co-transfected with plasmids for the expression of mCherry-BCL-2 (lower panels) before fixation after 18 h, mounting on slides using DAPI-containing mounting medium and cLSM. (H) TMD sequence is essential for membrane insertion of BOK and co-localization with BCL-2. MCF-7 cells were transfected with plasmids for the expression of mCherry-BOK$^{\Delta TMD}$ in combination with EYFP-Mito (upper panel) or EGFP-BCL-2 (lower panel). After 18 h, cells were fixed and mounted on slides using DAPI-containing mounting medium followed by cLSM. (C–H) Images are maximum projections of z-stacks representative of two independent experiments. Scale bar = 10 μm. Source data are available online for this figure.

because the RMSD of larger proteins/protein assemblies is intrinsically higher than that of smaller protein structures (Irving et al, 2001).

Taken together, our multiscaling MD simulations show that BOK-TMD and BCL-2-TMD interact dynamically in the ER membrane by multiple interaction interfaces. The requirements for interaction are best met in 2x BOK-TMD/2x BCL-2-TMD tetramers, which is consistent with the experimentally observed optimal 1:1 ratio of the interacting BAX-TMDs in the split luciferase assay.

### Peptide–lipid interaction

The here generated ER-like membrane mimic consisting of charged 1-palmitoyl-2-oleoyl-sn-glycero-3-phosphoinositol (POPI), zwitterionic 1-myristoyl-2-oleoyl-sn-glycero-3-phosphocholine (MOPC) and 1-myristoyl-2-oleoyl-sn-glycero-3-phosphoethanolamine (MOPE) and uncharged 1-palmitoyl-2-oleoyl-sn-glycerol (PODG) and cholesterol allows analysis of the protein-lipid interaction, which often is a strong modulator of protein-protein interactions (Pluhackova et al, 2016; Friess et al, 2018)—also for Bcl-2 proteins (Lutter et al, 2000; Lucken-Ardjomande et al, 2008; Shamas-Din et al, 2015). Regardless of the oligomeric state of the TMDs, MD simulations in the ER-mimic show strong depletion of MOPC, slight depletion of cholesterol, slight enrichment of MOPE and strong enrichment of POPI for both TMDs (Fig. 5H; Appendix Fig. S11). PODG is neither depleted nor enriched in the vicinity of BCL-2-TMD and slightly enriched at BOK-TMD. The strong enrichment of negatively charged POPI likely results from the high amount of positively charged amino acids in the TMDs (BOK-TMD contains 3 x Arg, 1 x Lys, 1 x Glu, and 1 x Asp; BCL2 TMD comprises 2 x Lys and 1 x Asp, Appendix Tables S11 and 12).

### L223, V226, and I230 in BCL-2-TMD contribute to interaction with BOK-TMD

The analysis of interaction energy per residue from MD simulations and the visual inspection of simulated BOK-TMD/BCL-2-TMD oligomers (Fig. EV4A,B) identified one side of the BOK-TMD helix which contains predominantly large hydrophobic residues (L190$^{BOK}$, A193$^{BOK}$, L194$^{BOK}$, F197$^{BOK}$, L201$^{BOK}$, A204$^{BOK}$, and L208$^{BOK}$) and faces exclusively hydrophobic lipid tails in all clusters formed (Fig. EV4A). In line, these residues negligibly contribute to the interaction energy of the BOK-TMD/BCL-2-TMD dimers (Fig. EV4B). Also, the BCL-2-TMD exhibits a lipid facing and TMD-TMD interacting side. Consequently, specific residues constitute the interaction interface between BOK-TMD and BCL-2-TMD in all hetero-oligomers including dimers, trimers, and tetramers. The interacting/stabilizing

residue L223$^{BCL-2}$ as well as V226$^{BCL-2}$ and I230$^{BCL-2}$ that constitute a VI$_4$-interaction motif (Senes et al, 2000) in BCL-2 were chosen to experimentally validate their relevance for TMD-TMD interaction. To this end, we generated plasmids for the expression of BCL-2-TMDs with mutation of these interacting amino acids to alanine preserving the hydrophobic character of the BCL-2-TMD (Fig. 6A). Specifically, mutants BCL-2-TMD L223A V226A I230A (LVI/AAA), L223A V226A (LV/AA), V226A I230A (VI/AA), and I230A (I/A) were used in NanoBiT assays and cLSM experiments. The different BCL-2-TMDs were transiently expressed as mTurq2-fusion in MCF-7/EYFP-Mito and MCF-7/EYFP-ER cells (Fig. 6B). Subsequent cLSM imaging revealed unchanged ER localization for all mutant BCL-2-TMDs I/A, LV/AA, VI/AA and the triple mutation LVI/AAA.

After cLSM confirmed unchanged ER localization of mutant BCL-2-TMDs, the split luciferase assay was applied to analyze the influence of TMD mutations on BOK-TMD/BCL-2-TMD interaction (Fig. 6C). This independent set of experiments confirmed increased relative normalized luminescence for the combination BOK-TMD/BCL-2-TMD (0.47) compared to the BAX-TMD/TOM5-TMD negative control (0.09) verifying BOK-TMD/BCL-2-TMD interaction. Each analyzed mutation of SmBiT-BCL-2-TMD significantly reduced luminescence signal in combination with LgBiT-BOK-TMD (from 0.47 to 0.21 for I/A, LV/AA, VI/AA or 0.13 for triple mutation LVI/AAA). Thus, all selected mutations of contact residues diminished the interaction signal of BOK-TMD and BCL-2-TMD by more than half (68–90%). Subsequently, we assessed whether the respective mutations of BCL-2-TMD impact on inhibition of apoptosis induction by BOK. In SW982 cells, wild-type mCherry-BCL-2 effectively reduced proportion of Annexin-V-APC positive (APC+, i.e., apoptotic) cells resulting from EGFP-BOK overexpression (from 23.5% to 11.5% APC+ cells) to the level of EGFP-expressing negative control (11.4% APC+ cells). Strikingly, mutation of BCL-2-TMD reduced or even abolished inhibition of BOK overexpression-induced cell death by BCL-2 (WT: 11.5% compared to LVI/AAA: ~18% APC+ cells, LV/AA: 15.8% APC+ cells, VI/AA: 26.5% APC+ cells, I/A: ~18% APC+ cells; Fig. 6D). Concludingly, reduced interaction of BOK-TMD with mutant BCL-2-TMDs as well as impaired inhibition of BOK overexpression-induced cell death by TMD-mutant BCL-2 corroborate the interaction of BOK-TMD and BCL-2-TMD via the identified residues. Also, these results substantiate the hypothesis that BOK-TMD and BCL-2-TMD interaction is functionally relevant for apoptosis with multiple residues of the interaction interface of BOK-TMD and BCL-2-TMD contributing to interaction.

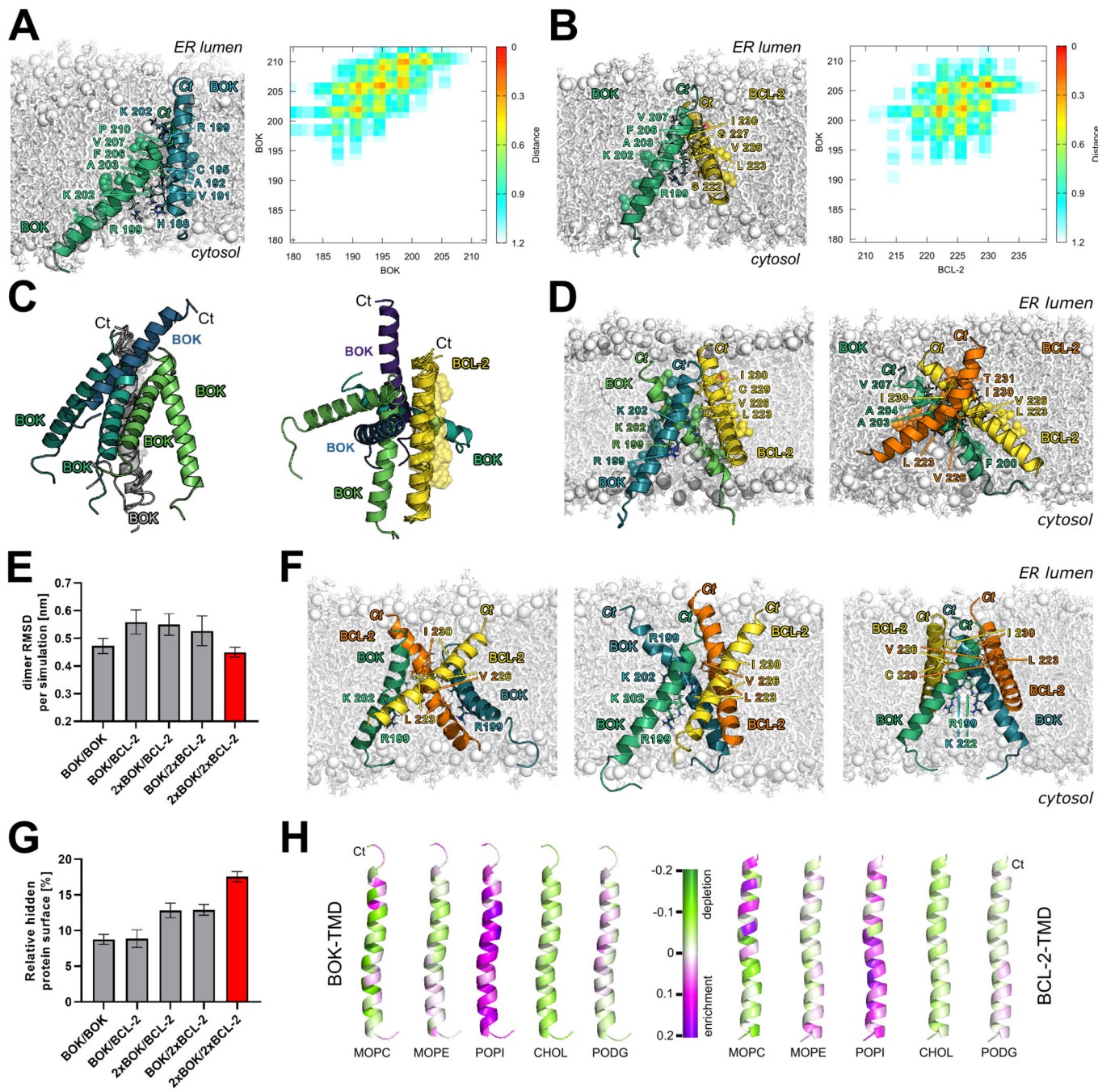

## BCL-2 antagonizes BOK-induced cell death TMD-dependently

BCL-2 is a bona fide antagonist of BAX and BAK and effectively counteracts BAX and BAK-induced mitochondrial pore formation and cell death (Ding et al, 2010; Willis et al, 2005). In contrast to BAX and BAK, cell death induction by BOK is less thoroughly understood. However, it is generally recognized that either BOK overexpression or blocked degradation induces apoptosis (Einsele-Scholz et al, 2016; Llambi et al, 2016). So far only few studies support a direct regulation of BOK-mediated apoptosis by anti-

apoptotic Bcl-2 proteins (Stehle et al, 2018; Lucendo et al, 2020). On the contrary, it is assumed that anti-apoptotic Bcl-2 proteins do not affect BOK-induced cell death (Carpio et al, 2015; Llambi et al, 2016).

While interaction of BOK with Bcl-2 proteins in general and BCL-2 in particular is disputed, the above results strongly suggest that BOK and BCL-2 interact via their TMDs. Consequently, we further investigated whether BCL-2 attenuates BOK-induced cell death in other cell lines. To this end, we performed siRNA-mediated knock-down of BCL-2 in HEK293 cells and subsequently transfected these cells to express EGFP-BOK. Expectedly, flow

◄   **Figure 5.   BOK-TMD and BCL-2-TMD interact in ER membranes.**

(A) BOK-TMD/BOK-TMD-I homodimer (55% of formed homodimers) exhibits an asymmetric interaction interface and causes membrane indentation of both membrane leaflets by snorkeling of R199[BOK] and K202[BOK] to the lipid headgroups. Snapshot after 1 μs AA simulation (left) and corresponding contact map (right). (B) BOK-TMD/ BCL2-TMD-I heterodimer (44% of formed heterodimers) is crossed shaped and stabilized by the interaction interface formed by the C-terminal halves of the TMDs. Positively charged residues of both peptides, i.e., R199[BOK], K202[BOK], and K218[BCL-2] snorkel to the lipid headgroups of the cytoplasmic membrane leaflet only and cause a significant local thinning of the bilayer. Snapshot after 1 μs AA simulation (left) and corresponding contact map (right). (C) The interaction partners bind to a reference TMD (BOK-TMD in gray in case of BOK-TMD/BOK-TMD homodimers on the left and BCL-2-TMD in yellow for BOK-TMD/BCL2-TMD heterodimers on the right) in diverse positions, suggesting a possibility to form higher-order oligomers. Lipid-facing residues of the reference TMD are shown as transparent spheres. (D) Representative snapshots after 1 μs AA simulations of the most often formed 2x BOK-TMD/BCL-2-TMD heterotrimers on the left and BOK-TMD/2x BCL-2-TMD heterotrimers on the right. (E) Average RMSD (estimated on a per dimer basis) values of AA simulations relative to the backmapped CG structure. The error bars denote SEM over individual simulations. $N = 13$ for BOK-TMD homodimers, $N = 15$ for BOK-TMD/BCL-2-TMD heterodimers, $N = 9$ and $N = 10$ for 2x BOK-TMD/BCL-2-TMD and BOK-TMD/2x BCL-2-TMD heterotrimers, respectively, and $N = 18$ for 2x BOK-TMD/2x BCL-2-TMD heterotetramers. Bar for heterotetramers showing lowest RMSD is colored red. (F) Stable 2x BOK-TMD/2x BCL-2-TMD heterotetramers with "twisted" compact shape after AA simulation. On the left and on the right, heterotetramers with symmetric BCL-2-TMD and BOK-TMD homodimer in the center, respectively, are shown. In the middle, a heterotetramer comprised of two BOK-TMDs followed by two BCL-2-TMDs is visualized. All tetramers are right-handed. (G) Protein surface occupied by protein-protein interactions relative to the total surface of all peptides for all simulation types. The error bars denote SEM and the number of simulations equals to that listed in (E). (H) Lipid enrichment (magenta) or depletion (green) per BOK-TMD (left) or BCL-2-TMD (right) residue in the heterotetramer simulations. Data information: In (A), (B), (D), and (F) water and ions were omitted for clarity, lipids are shown as white sticks with phosphorus atoms highlighted as spheres. The proteins are shown as cartoons with selected amino acid sidechains shown as sticks with label. BOK-TMDs are colored green or teal BCL-2-TMDs are colored yellow or orange. Source data are available online for this figure.

cytometric analysis showed that overexpression of BOK substantially increased the proportion of APC+ cells to almost half of transfected (EGFP positive) cells (37.9% APC+ cells) 18 h post transfection (Fig. 7A). Knock down of BCL-2 resulted in an increased proportion of apoptotic cells as compared to control cells (49.7% APC+ cells) indeed indicating an antagonistic role of BCL-2 in BOK overexpression-induced cell death. Because knock down reduced but did not abolish BCL-2 expression as analyzed by Western blot (Fig. EV5B) we aimed to verify BCL-2-mediated inhibition of BOK-induced apoptosis by CRISPR/Cas9-mediated knock-out. We co-transfected HEK293 to express CRISPR/Cas9 for BCL2 knock-out together with EGFP-BOK and subsequently analyzed apoptosis induction in EGFP-positive cells by Annexin-V-APC staining. Flow cytometric analysis detected 50.2% apoptotic EGFP-BOK-expressing cells which was enhanced to 60.6% in BCL-2 knock-out cells (Fig. EV5C). Next, we analyzed BOK-induced apoptosis in the BAX- and BCL-2-deficient cell line DU145 (Fig. EV5A). Overexpression of EGFP-BOK induced exposure of phosphatidyl serine in 12.3% of DU145 cells after 18 h and 20.7% of DU145 cells 42 h post transfection as assessed using flow cytometry (Figs. 7B and EV5D). Of note, we observed that overexpression of BOK was generally less efficient than overexpression of other Bcl-2 proteins (Fig. 7D). Thus, functional implication of BCL-2 in BOK-induced cell death could only be analyzed for the small fraction of cells expressing sufficient amounts of EGFP-BOK to induce cell death. However, apoptosis induction by BOK was efficiently reduced by co-transfection of mCherry-BCL-2 resulting in 5.7% APC+ (compared to 12.3%) cells after 18 h and 8.2% APC+ (compared to 20.7%) cells 42 h post transfection (Figs. 7B and EV5D). In line with apoptosis induction, cleavage of the caspase substrate PARP indicating apoptosis execution was readily detected after EGFP-BOK overexpression which was reduced by co-transfection with mCherry-BCL-2 (Fig. 7C,D). Moreover, assessing activity of apoptosis-specific caspase-3/7 in EGFP-BOK-expressing DU145 cells we detected an increase of 32% in caspase-3/7 activity compared to control cells while caspase-3/7 activity was decreased by 17% by co-expression of BCL-2 compared to control (Fig. 7E).

With the BAX and BCL-2 deficient cell line DU145, we sought to further confirm whether the interaction of BOK-TMD and BCL-2-

TMD is relevant for the observed cell death inhibition. Since the BAX/ BCL-2 co-localization seems to be mediated by BH3 domain:hydrophobic groove rather than depend on their TMDs, we analyzed whether cell death induction by chimeric BOK with BAX-TMD is reduced by co-expression of BCL-2. The exchange of BOK-TMD to BAX-TMD shifted localization of BOK to mitochondria (Fig. EV5G) and also potentiated cell death induction by BOK overexpression from 12.3% to 20.0% APC+ cells 18 h post transfection (Fig. 7B). Manifesting a clear role of TMD interaction, cell death induction by BOK[BAX-TMD] was not altered by co-expression of BCL-2 (18.5% APC+ cells). Failure of BCL-2 to inhibit BOK[BAX-TMD] induced cell death was also reflected by unchanged level of cleaved PARP and caspase-3/7 activity after expression of EGFP-BOK[BAX-TMD] alone compared to co-expression with mCherry-BCL-2 (Fig. 7C–E).

Interestingly, Western blot analysis showed higher expression of mitochondria-localized EGFP-BOK[BAX-TMD] compared to ER-localized wild-type EGFP-BOK (Fig. 7C). Densitometric analysis of cleaved PARP per EGFP suggests that, although showing a higher expression level, BOK[BAX-TMD] induces apoptosis less efficiently than wild-type BOK (rel. intensity cPARP/EGFP-BOK[BAX-TMD] = 17.5; rel. intensity cPARP/EGFP-BOK = 60.5, Fig. 7D). The increased pro-apoptotic activity of BOK with BAX-TMD hence may result from higher expression due to increased stability or reduced degradation compared to wild-type BOK.

To further challenge the TMD dependency of BCL-2-mediated inhibition of BOK-induced apoptosis, we analyzed cell death induction by BOK carrying the wild-type TMD or the TMD of cb5 in the presence or absence of BCL-2. As for BOK[BAX-TMD], the low cell death induction by BOK[cb5-TMD], (7.7% APC+ cells), was not reduced by BCL-2 (7.4% APC+ cells, Fig. 7F). Vice-versa, exchanging the TMD of BCL-2 to the TMD of MOM-localized protein TOM5 reduces its calculated inhibitory capacity towards BOK-induced apoptosis when co-transfected in DU145 cells by 67% (cell death increases from 5.1% APC+ cells to 8.0% APC+ cells, Fig. 7G).

In conclusion, knock-down or knock-out of BCL-2 increased BOK-induced apoptosis, while co-transfection with BCL-2 reduced cell death induction by overexpression of BOK. Thereby the BCL-2-mediated inhibition of BOK-induced apoptosis specifically depends

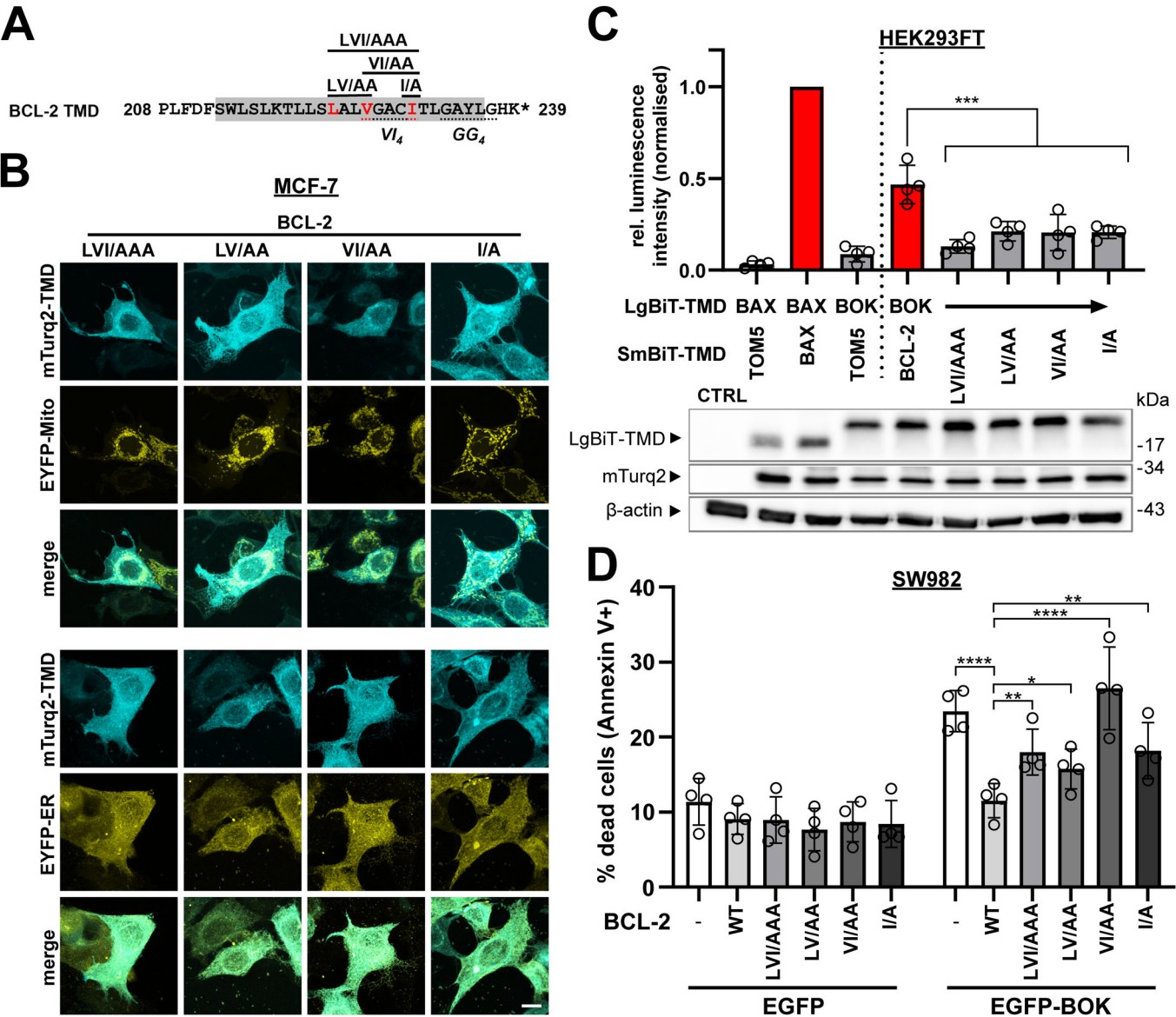

**Figure 6. Hydrophobic residues L223, V226, and I230 in BCL-2-TMD mediate interaction with BOK-TMD.**

(A) Sequence of BCL-2-TMD with underlined interaction motifs $GG_4$ and $VI_4$. Residues in the interaction interface that were mutated are highlighted in red. Generated mutants are shown on top. The gray shade indicates the α9 helix in BCL-2. (B) Mutated TMD peptides of BCL-2 are still localized to ER membranes. MCF-7 cells stably expressing EYFP-Mito or EYFP-ER were transfected with plasmids for the expression of mTurquoise2 (mTurq2)-labeled BCL-2-TMD peptides carrying indicated mutations. Cells were fixed after 24 h and TMD localization was analyzed by cLSM. Images are maximum projection of z-stacks representative of two independent experiments. Scale bar = 10 μm. (C) Mutation of BCL-2-TMD residues in the interaction interface with BOK-TMD abrogates their interaction. Split luciferase assay in HEK293FT cells transfected with plasmids for the expression of indicated NanoBiT-TMDs. Cells were harvested after 24 h and samples were both used for split luciferase assay and Western blot. BAX-TMD/TOM5-TMD serves as a negative control, while BAX-TMD/BAX-TMD serves as positive control. Mutants of SmBiT-BCL-2-TMD were combined with LgBiT-BOK-TMD. Graphs show luminescence intensity relative to the positive control and normalized to mTurquoise2 fluorescence intensity. Shown is the mean ± sd from four independent experiments. In Western blots (bottom), corresponding whole cell lysates were analyzed for expression of LgBiT and mTurquoise2 (mTurq2). Representative blot from three independent experiments is shown. ***$p < 0.001$, one-way ANOVA with Dunett's multiple comparison test. (D) TMD mutation reduces BCL-2's ability to inhibit BOK-induced cell death. SW982 cells were transfected with plasmids for the co-expression of EGFP or EGFP-BOK with mCherry-BCL-2 wild-type, mutants or empty vector as a control. After 24 h, cells were stained with Annexin V-APC and EGFP + /Annexin-V+ cells were detected by flow cytometry. Shown is the mean ± sd from four independent experiments. *$p < 0.05$, **$p < 0.01$, ****$p < 0.0001$, one-way ANOVA with Dunett's multiple comparison test. Source data are available online for this figure.

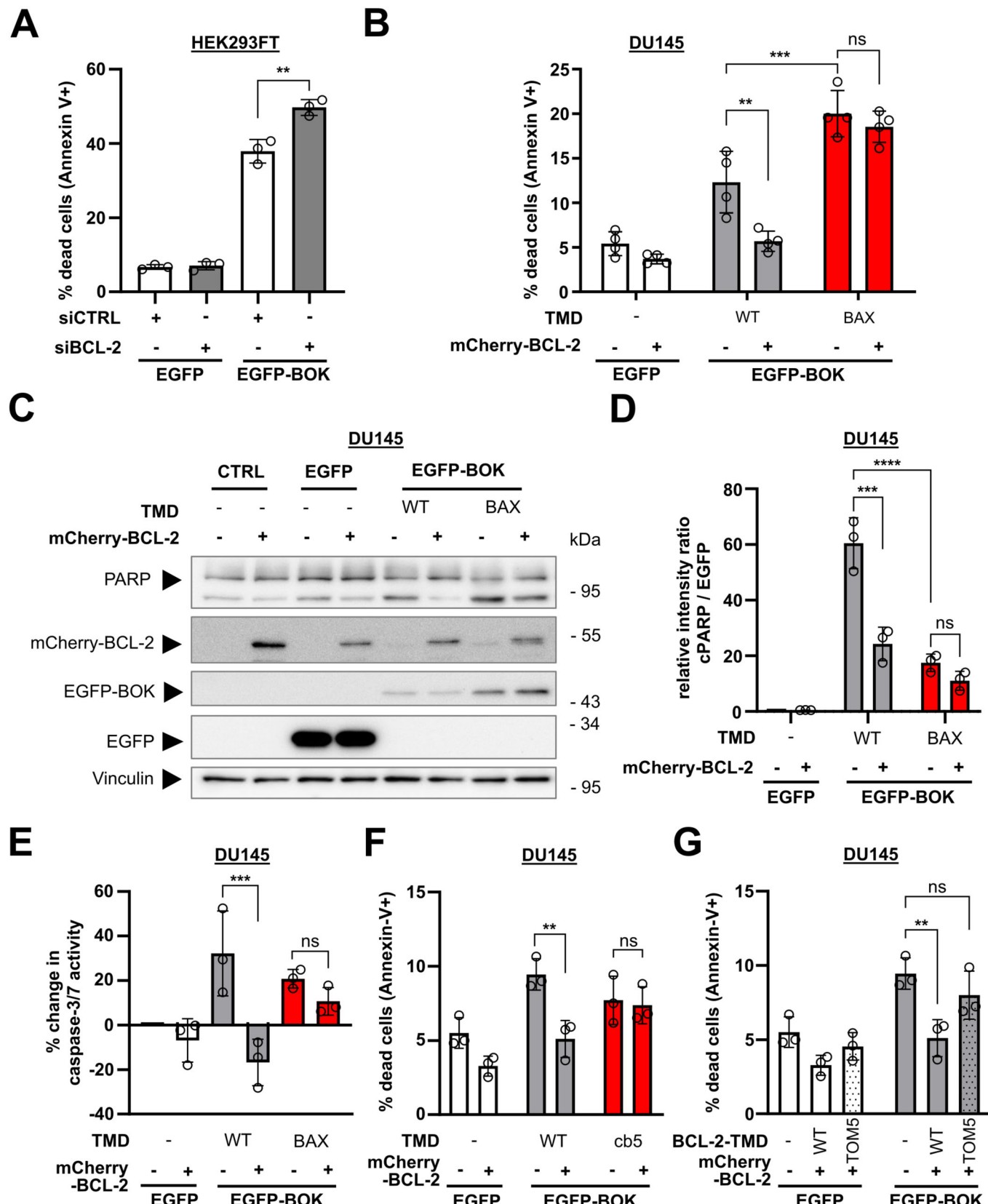

◄ **Figure 7. Inhibition of BOK-mediated apoptosis by BCL-2 depends on BOK-TMD.**

(A) BOK-induced apoptosis is pronounced after knock-down of BCL-2. HEK293FT were transfected with non-targeted (siCTRL) or BCL-2-targeted (siBCL-2) small-interfering RNA for 24 h and subsequently with plasmids for the expression of EGFP or EGFP-BOK for additional 18 h. Cells were stained with Annexin-V-APC and analyzed by flow cytometry. (B) Co-expression of mCherry-BCL-2 reduces apoptosis induction by BOK but not BOK$^{BAX-TMD}$. DU145 cells were transfected with plasmids for the expression of EGFP, EGFP-BOK, or EGFP-BOK$^{BAX-TMD}$ without or with mCherry-BCL-2. After 18 h, cells were stained with Annexin V-APC and EGFP+/Annexin-V+ cells were detected by flow cytometry. (C) Expression of BCL-2 reduces abundance of cleaved PARP in BOK-expressing cells but not in BOK$^{BAX-TMD}$ expressing cells. DU145 cells expressing EGFP, EGFP-BOK, or EGFP-BOK$^{BAX-TMD}$ alone or in combination mCherry-BCL-2 were analyzed by Western blot for the expression of EGFP(-BOK), BCL-2, and PARP. Vinculin served as loading control. Representative blot from three independent experiments. (D) Densitometric analysis of three independent experiments as shown in (C). (E) BOK-induced caspase-3/7 activity is reduced by BCL-2, but not for BOK$^{BAX-TMD}$. DU145 cells were co-transfected with plasmids for the expression of EGFP, EGFP-BOK, or EGFP-BOK$^{BAX-TMD}$ with or without mCherry-BCL-2. After 18 h, caspase-3/7 activity was assessed with the Caspase-Glo 3/7 assay kit (Promega). Percent change in caspase-3/7 activity compared to EGFP control is shown. (F) mCherry-BCL-2 reduces apoptosis induction by BOK but not BOK$^{cb5-TMD}$. DU145 cells were transfected with plasmids for the expression of EGFP, EGFP-BOK, or EGFP-BOK$^{cb5-TMD}$ without or with mCherry-BCL-2. After 18 h, cells were stained with Annexin-V-APC and EGFP+/Annexin-V+ cells were detected by flow cytometry. (G) Inhibition of BOK-induced cell death by BCL-2 is impaired for BCL-2$^{TOM5-TMD}$. DU145 cells were transfected with plasmids for the expression of EGFP or EGFP-BOK in combination with mCherry-BCL-2 or mCherry-BCL-2$^{TOM5-TMD}$. After 18 h, cells were stained with Annexin-V-APC and EGFP + /Annexin-V+ cells were detected by flow cytometry. Data information: Graphs show mean ± sd from three (A, E, F, G, H) or four (B, C) independent experiment. **$p < 0.01$, ***$p < 0.001$, ****$p < 0.0001$, one-way ANOVA with Tukey's multiple comparison test. Source data are available online for this figure.

on their TMD. Thus, to the best of our knowledge, for the first time our data uncover a functional involvement of BOK-TMD and BCL-2-TMD interaction in apoptosis signaling.

## Discussion

BH3 mimetics which made it possible to target interactions of Bcl-2 proteins equipped humanity with new powerful options to stand against cancer. BCL-2 as one of the most well-studied anti-apoptotic family members was shown to be a worthwhile target for anti-cancer therapy: cancer cell-protective BCL-2 overexpression and hence sequestration of pro-apoptotic proteins is a driving force of tumorigenesis in hematopoietic malignancies like chronic lymphoid leukemia (CLL) (Robertson et al, 1996; Del Gaizo Moore et al, 2007). The identification of BCL-2 as an oncogene ignited the development of BH3 mimetics and eventually the specific BCL-2 inhibitor ABT-199/Venetoclax. Venetoclax was the first FDA-approved BH3 mimetic for the use as a single agent in CLL and later in combination with azacidine, decitabine or low dose cytarabine in acute myeloid leukemia (AML) (Roberts et al, 2016; DiNardo et al, 2019).

Importantly, the Bcl-2 protein family is furthermore a valuable target for anti-cancer therapy. Recent studies revealed that Venetoclax in addition to potently blocking BCL-2 causes cellular responses that support anti-tumor activity including metabolic reprogramming and activation of the integrated stress response (Roca-Portoles et al, 2020; Weller et al, 2022). Undoubtedly, identifying new target structures within the Bcl-2 family will aid in the endeavor to develop even more specific treatment options.

Here, we established a bimolecular split luciferase assay and elucidated a novel interaction of BCL-2 and BOK via their TMDs. The presented data furthermore indicates functional relevance of the TMD interaction in apoptosis regulation. Surprisingly, the interactions of BCL-2-TMD and BOK-TMD takes place at ER membranes and MD simulations in an ER membrane mimic revealed formation of stable hetero-tetramers (2x BOK-TMD/2x BCL-2-TMD), structurally dynamic BOK-TMD homodimers, BCL-2-TMD/BOK-TMD hetero-dimers and hetero-trimers. Also, MD simulations identified decisive BCL-2-TMD residues for BOK-TMD/BCL-2-TMD interaction that were confirmed by using respective mutant BCL-2-TMDs in the split luciferase assay and

co-expressing mutant full-length BCL-2 with BOK. Further in vitro experiments support a functional significance of the BCL-2-TMD interaction with BOK-TMD as BCL-2-mediated inhibition of BOK-induced cell death was clearly dependent on BOK-TMD sequence.

The newly described interaction of BOK-TMD with BCL-2-TMD is particularly intriguing since the interaction of BOK via its BH3 domain is still debated. In fact, specific BH3 domain interaction of BOK with anti-apoptotic Bcl-2 proteins has not yet been reported. However, accompanying the identification of BOK a yeast two-hybrid assay found MCL-1 and A1 as BOK interaction partners (Hsu et al, 1997). Interaction of BOK and MCL-1 was also confirmed by other groups (Llambi et al, 2016; Stehle et al, 2018). More recently, interaction of BOK with MCL-1 via the TMD domain was shown to modulate apoptosis (Lucendo et al, 2020). In the same study, Lucendo et al show that MCL-1-TMD can tether BOK-TMD to mitochondria and increase the number of mitochondria-associated membranes (MAMs). Consequently, interaction via the TMD appears to be especially important in case of BOK as compared to other Bcl-2 proteins. However, in the present study MCL-1-TMD was localized to the ER rather than mitochondria and the split luciferase assay did not indicate interaction with BOK-TMD. In support of the importance of BOK-TMD for protein-protein interaction, abrogation of BH3-mediated interaction by mutation of the conserved L70 in BOK did not alter the co-localization with BCL-2.

Also the mode of BOK-induced apoptosis is controversial. Some reports claim BOK to be a pro-apoptotic effector like BAX and BAK which kill cells by forming pores in the MOM (Einsele-Scholz et al, 2016; Fernández-Marrero et al, 2017; Shalaby et al, 2023). Other studies deny BOK any independent apoptosis-mediating function and suggest that BOK functions upstream of BAX and BAK (Echeverry et al, 2013; Carpio et al, 2015). A solution to this conundrum might be offered by recent reports showing that BOK localizes to the ER and is involved in calcium transfer to mitochondria via mitochondria-associated membranes (MAMs) thereby promoting MOM depolarization and apoptosis (Carpio et al, 2021). Hand in hand with BOK localization at the ER, a role of BOK in ER stress response has also been reported (Echeverry et al, 2013; Carpio et al, 2015; Walter et al, 2022). In concordance, in the present study BOK and BCL-2 co-localize at the ER suggesting a regulatory role of their TMD interaction at the ER or MAMs. In line with ER localization, BOK and BCL-2 both interact with IP3R

calcium channels with opposing effects on apoptosis (Rong et al, 2009; Monaco et al, 2012; Schulman et al, 2016). In fact, for BCL-2 the TMD is necessary and sufficient to bind and inhibit IP3R (Ivanova et al, 2016). However, BCL-2 binds IP3Rs also via the BH4 motif to promote cell survival and decrease apoptosis by mediating $Ca^{2+}$ leakage from the ER to the cytosol and inhibiting $Ca^{2+}$ release upon apoptosis induction (Rong et al, 2009; Monaco et al, 2012). On the other hand, stability as well as pro-apoptotic capacity of BOK largely depend on the binding to IP3R as Schulman et al demonstrated that virtually all cellular BOK is bound to IP3R and rapidly degraded via the proteasome when released (Schulman et al, 2016). The authors furthermore revealed that BOK deletion leads to fragmentation of mitochondria and thereby BOK affects bioenergetics suggesting further apoptosis-unrelated functions of BOK (Schulman et al, 2019). The opposing role of BOK and BCL-2 in apoptosis by binding to IP3Rs and their function in $Ca^{2+}$ signaling indicate ER and MAMs as the functional hub for their TMD interaction, for example, by competing for binding sites at IP3Rs and, hence, regulation of IP3R function.

The function of BOK and BCL-2 in $Ca^{2+}$ signaling at the ER does not exclude an involvement of TMD interaction in canonical pore formation by BOK in the MOM since Bcl-2 family proteins that mainly reside at the ER such as BCL-2 and the BH3-only protein BIK, translocate to mitochondria in order to interact with mitochondrial localized Bcl-2 family proteins and regulate MOMP (Lalier et al, 2021; Osterlund et al, 2023). The here presented high-throughput MD simulations of BOK-TMD and BCL-2-TMD in an ER membrane mimic show that BCL-2-TMD binds to BOK-TMD and preferentially forms higher order oligomers. Interestingly, in these simulations BCL-2-TMD or BCL-2-TMD homodimers associate with BOK-TMD homodimers which indicates that BCL-2 is recruited to BOK homodimers and inhibits BOK oligomerization. Along these lines, although heterodimers and heterotrimers showed high transformability, the tetramers consisting of two BOK-TMD and two BCL-2-TMD peptides showed the highest structural stability among all oligomers studied, further supporting a role of TMD interaction in BOK oligomerization. Additionally, snorkeling of positively charged lysine and arginine of BOK to the lipid headgroups resulted in local membrane indentation demonstrating the capacity of BOK-TMD to modulate membrane structure. Vice versa, membrane composition likely is a major factor that influences TMD interaction, since negatively charged POPI accumulates in the vicinity of both BOK-TMD and BCL-2-TMD. Enrichment of negatively charged POPI at TMDs is in line with reports showing that negatively charged cardiolipin (CL) in the MOM significantly influences interaction of Bcl-2 proteins with the mitochondrial membrane (Lutter et al, 2000; Lucken-Ardjomande et al, 2008; Shamas-Din et al, 2015). CL interacts with positively charged amino acids in TMDs of which some can be found in BOK and BCL-2 as well (R199[BOK], K202[BOK], and K218[BCL-2]). In MD simulations, these charged residues also attracted negatively charged POPI lipids. Several defined motifs which contain hydrophobic amino acids like glycine, alanine, leucine/isoleucine, and valine are frequently involved in helix-helix interactions and TMD oligomerization (Russ and Engelman, 2000; Senes et al, 2000; Gössweiner-Mohr et al, 2022). In agreement, we identified specific hydrophobic residues constituting the interaction interface of BOK-TMD and BCL-2-TMD. Remarkably, a VI[4] interaction motif is found in the BCL-2-TMD (Fig. 6A,

amino acids 226–230) located at the identified TMD-TMD interface. Mutation of BCL-2-TMD at L223[BCL-2] next to the VI[4] as well as V226[BCL-2] and I230[BCL-2] constituting the VI[4] abrogated interaction with BOK-TMD. Although BCL-2-TMD carries another interaction motif, the GG[4] motif (amino acid 233–237), the VI[4] motif apparently is the dominant interaction-defining element in the BCL-2-TMD. Intriguingly, none of these frequent motifs could be found in BOK-TMD. However, the sequence of BOK-TMD contains a peculiar high number of phenylalanine residues (Fig. EV4A, F197[BOK], F200[BOK], F205[BOK], F206[BOK]). This is especially interesting with regard to an interaction-promoting effect of phenylalanine in TMDs (Unterreitmeier et al, 2007; Kwon et al, 2016). Thus, studying further motifs and residues in the BCL-2-TMD and BOK-TMD by in silico approximations and in vitro mutation will grant further insight into their role in the interaction of BOK-TMD with BCL-2-TMD.

Also post-translational modification of Bcl-2 TMDs strongly impact localization and function of Bcl-2 proteins, exemplified by phosphorylation of the BAX-TMD at S184 by Akt (Guedes et al, 2013; Kale et al, 2018a) that prevents integration into the MOM. Intriguingly, the TMD mutation G179E in BAX promotes resistance to anti-cancer treatment by rendering cells less susceptible to Venetoclax (Fresquet et al, 2014). Thus, mutation and post-translational modification of the TMD in Bcl-2 proteins are highly relevant for apoptosis regulation. In addition, as shown for the BH3-only proteins BIML as well as PUMA, high-affinity binding to anti-apoptotic Bcl-2 proteins depends on a "double-bolt lock" mechanism mediated by both TMD and BH3 domain (Liu et al, 2019; Pemberton et al, 2023). Hence, these results might serve as a role model for a new generation of Bcl-2 protein-targeting drugs that target interaction not only via BH3 domain:hydrophobic groove but also the TMD. Interestingly, we found several tumor-specific TMD mutations in the COSMIC database (cancer.sanger.-ac.uk/cosmic) underpinning the impact of Bcl-2 TMDs on the apoptosis signaling network by affecting localization, interaction or "double-bolt lock" mechanism.

Taken together, we unveil two new aspects of apoptosis regulation: Firstly, we have discovered that the inhibitory role of BCL-2 on BOK-mediated apoptosis crucially depends on the interaction via their TMDs. Secondly, BOK and BCL-2 are localized at the ER via their TMDs which indicates that BOK-induced apoptosis regulation originates at the ER, not the mitochondria. Our findings support the emerging role of TMD interactions as an important structural component of the Bcl-2 interaction network. We conclude that apoptosis dysregulation as well as survival strategies in healthy and transformed cells are inherently influenced by Bcl-2 TMDs in two ways: (i) subcellular localization of Bcl-2 proteins and (ii) interaction of specific Bcl-2 protein via their TMDs. Simultaneous targeting of BH3 domain:hydrophobic groove and the TMD interface of Bcl-2 proteins, therefore, harbors not yet exploited potential for anti-cancer therapy, for example, by increasing drug specificity.

## Methods

### Protein sequences

Protein names, function and Uniprot entries (www.uniprot.org) of protein sequences used for generation of expression vectors are

listed in Appendix Table S1. TMD sequences used are listed in Appendix Table S2.

## Antibodies and reagents

Antibodies used were: anti-LgBiT (Promega #N7100), anti-GFP (for EGFP/mTurq2, Santa Cruz #sc-9996), anti-GAPDH (Cell Signaling Technology (CST) #2118), anti-BOK (abcam ab186745), anti-BCL-2 (CST #15071), anti-PARP (CST #9542), anti-Vinculin (protein tech #66305-1-Ig), anti-MCL-1 (CST #5453), anti-BCL-XL (CST #2762), anti-BAX (CST #2772), anti-BAK (CST #3814), anti-β-actin (Sigma #A5316), secondary horseradish peroxidase-conjugated antibodies anti-mouse (CST #7076) and anti-rabbit (CST #7074); Vectors for expression of EGFP, EYFP-Mito (EYFP fused to mitochondrial targeting sequence of cytochrome C oxidase subunit VIII) and EYFP-ER (EYFP fused to Calreticulin ER-targeting sequence) were from Clontech (Takara Bio, Kusatsu, Japan); Vector for CRISPR/Cas9-mediated knock-out of BCL-2 was plentiCTRSPRv2 (Addgene #52961) with gRNA targeting exon 1 in BCL2; reagents for cell stainings and fixation used were Mitotracker green/red CMXRos (Thermo Fisher Scientific, Waltham, MA, USA), fixation solution ROTI Histofix 4% formaldehyde (Carl Roth, Karlsruhe, Germany), ProLong Diamond Antifade Mountant with DAPI (Thermo Fisher Scientific). Nuclear and ER staining for high-content analysis was done with DRAQ5 and BODIPY-thapsigargin (ThermoFisher Scientific).

## Cell culture

Human embryonic kidney cell line HEK293FT and breast cancer cell line MCF-7 were cultivated in Roswell Park Memorial Institute medium (RPMI, Thermo Fisher Scientific, Gibco) supplemented with 10% FCS (fetal calf serum, Merck, Darmstadt, Germany) and 1% penicillin/streptomycin (Thermo Fisher Scientific, Gibco). Prostate carcinoma cell line DU145 and SW982 synovial sarcoma cell line were cultivated in Dulbecco's modified Eagle's medium (DMEM, Thermo Fisher Scientific, Gibco) supplemented with 10% FCS and 1% Penicillin/Streptomycin. All cell lines were maintained at 37 °C and 5% $CO_2$ and regularly authenticated via STR-profiling (Eurofins Genomics Germany, Ebersberg, Germany). For harvesting and seeding, cells were detached using 0.05% Trypsin-EDTA (Thermo Fisher Scientific, Gibco) and processed as described.

## Transgene expression

Cells were transfected with the indicated plasmids using PEI (polyethylenimine hydrochloride (PEI MAX 40K), Polysciences, Warrington, PA, USA) according to the manufacturer's protocol. Samples used for microscopy were supplemented with 10 μM pan-caspase inhibitor Q-VD-OPh (Sellekchem, Houston, TX, USA). If not stated otherwise, transfected cells were kept at 37 °C and 5% $CO_2$ for 18 h and subsequently harvested for indicated experiments.

## siRNA-mediated knock-down

Knock-down experiments were performed as described previously (Weller et al, 2022). Briefly, cells were seeded 24 h prior transfection and then transfected with ON-TARGET Plus Smart-pool siRNA targeting BCL2 (siBCL-2) or non-targeted (siCTRL,

Horizon Discovery, Waterbeach, UK) using DharmaFECT 1 reagent (Horizon Discovery) according to the manufacturer's protocol. Twenty-four hours after knock-down cells were transfected with expression vectors encoding for EGFP or EGFP-BOK and incubated for another 18 h followed by flow cytometric analysis. For validation of knock-down efficiency cell lysates of transfected cells were analyzed by Western blot.

## Generation of vectors for transgene expression

Plasmids encoding for NanoBiT TMD fusion proteins were cloned using the NEBuilder system (New England Biolabs, Ipswich, MA, USA). Promega NanoBiT plasmid backbone (pBiT_1.1-C[TK/LgBiT], Promega, Madison, WI, USA) was combined with overlapping fragments generated by PCR which encoded for (i) either mCitrine or mTurquoise2, (ii) T2A sequence (iii) LgBiT or SmBiT connected by a hydrophilic linker with (iv) TMD sequences (TMD and primer sequences in Appendix Tables S1 and S2). T2A-SmBiT-TOM5-TMD fragment was generated via synthesis (GeneArt, Thermo Fisher Scientific). Plasmids encoding for fluorescent TMD probes were cloned likewise combining the pEGFP-C1 backbone (Clontech, Kyoto, Japan) with overlapping fragments generated by PCR which encoded for mTurquoise2 and TMD sequences from the previously generated NanoBiT vectors (Appendix Table S3). Plasmids encoding for TMD chimeras were cloned likewise combining Fluorophore-fused protein-encoding plasmids containing protein cores (BOK/BCL-2) described previously (Einsele-Scholz et al, 2016) with overlapping fragments generated by PCR (Appendix Table S4) which encoded for TMD sequences (BAX/cb5/TOM5). Plasmid inserts were verified by sequencing.

## Site-directed mutagenesis

Mutations were introduced in vectors using the Phusion site-directed mutagenesis kit (Thermo Fisher Scientific, Waltham, MA, USA). In brief, methylated template cDNA and mutation primers (Appendix Table S5) were mixed with reaction buffer on ice and Phusion high-fidelity DNA polymerase was added. Vectors with mutation were then produced in a PCR reaction and template DNA was digested by addition of DpnI restriction endonuclease.

## Flow cytometry

Cells and supernatant were harvested, cells were pelleted and washed in ice-cold PBS. Cell pellets were resuspended in 300 μl FACS PBS (PBS + 2% FCS) and if cells were not stained otherwise fluorescence was analyzed using a FACS Lyric flow cytometer (Becton Dickinson, Franklin Lakes, NJ, USA). Annexin-V-APC staining: After washing in PBS, cells were resuspended in 300 μl recombinant chicken Annexin-V-APC (ImmunoTools, Friesoythe, Germany) diluted 1:200 in Annexin-V-binding buffer (PBS, 2.5 mM $CaCl_2$). Samples were incubated on ice for 10 min and analyzed using FACS Lyric (BD).

## Caspase-3/7 activity

Cells were seeded in 12-well plates (100,000 cells/well) and incubated at 37 °C over night. The next day, cells were transfected with indicated plasmids and incubated for another 18 h. Then, cells

were trypsinized and collected together with floating cells in supernatant medium. After centrifugation ($500 \times g$, 5 min), cells were washed with PBS once and distributed to three wells of a white-bottom 96-well plate (Thermo Fisher Scientific). Caspase-Glo 3/7 luciferase substrate (Promega) was added in a 1:1 ratio to each well followed by gentle shaking for 30 s. After 30 min incubation at RT, luminescence was detected with a Victor Nivo multimode plate reader (PerkinElmer, Waltham, MA, USA).

## Western blot

Western blot was performed as described previously (Weller et al, 2022). In short, cells were harvested and cell pellets were resuspended in lysis buffer (50 mM Tris-HCl pH 7.6, 250 mM NaCl, 0.1% Triton X-100, 5 mM EDTA) supplemented with protease and phosphatase inhibitor cocktails. After sonication and clearance by centrifugation (15 min, $14,000 \times g$, 4 °C) relative protein content was analyzed and cell lysates were diluted and incubated in Laemmli buffer (5 min, 95 °C). Samples were then separated by SDS-PAGE and semi-dry blotted onto nitrocellulose membrane. After blocking with 5% skim milk powder in TBS-T (TBS, 0.1% Tween-20) for 1 h, primary antibodies were applied in 5% BSA or 5% skim milk powder in TBS-T and incubated over night at 4 °C. Blots were washed thrice in TBS-T for 10 min and afterwards HRP-coupled secondary antibodies diluted 1:2000 in 5% skim milk powder were applied for 1 h. TBS-T wash was repeated thrice for 10 min, ECL solution (SuperSignal West Dura, Thermo Fisher Scientific) was applied for 5 min and bands were detected using a STELLA imaging system (Raytest Isotopenmessgeräte GmbH, Straubenhardt, Germany).

## Generation of BMK/DKO cell lines

Baby mouse kidney $BAX^{-/-}/BAK^{-/-}$ double knockout cells (BMK/DKO) (Degenhardt et al, 2002) were transduced with lentiviral particles (pLVX-EF1a-puro) for the expression of mTurquoise2-fused TMDs or were transfected to express the full-length mCerulean3-fused BIK as described previously (Osterlund et al, 2023). Individual clones with detectable expression were isolated by flow cytometry and clonal cell lines with similar fluorescence intensity were selected.

## Confocal laser scanning microscopy (cLSM)

Cells were grown on coverslips (#1, Paul Marienfeld. Lauda-Königshofen, Germany), transfected with indicated plasmids and subsequently fixed with 4% paraformaldehyde solution (ROTI Histofix, Carl Roth) for 20 min at 37 °C. Then, cells were washed thrice with PBS and coverslips were mounted on microscopy slides (VWR, Radnor, PA, USA) with either DAKO fluorescent mounting medium (Agilent, Santa Clara, CA, USA) or ProLong Diamond antifade mountant with DAPI (Thermo Fisher Scientific).

### Mitrotracker staining
Cells were stained prior to fixation in 100 nM of Mitotracker Red CMXRos (Thermo Fisher Scientific) in unsupplemented culture medium for 5 min at 37 °C. Afterwards, cells were washed once in unsupplemented culture medium.

Images were acquired as z-stacks with a Leica TCS SP8 confocal laser-scanning microscope (Leica, Wetzlar, Germany) equipped with a

HC PL APO CS2 63x/1.40 oil immersion objective and excitation lasers at 405 nm, 488 nm, 552 nm, and 638 nm. Appropriate slit settings were used for detection of DAPI (415–480 nm), mTurquoise2 (415–480 nm), GFP (500–540 nm), mCitrine (500–540 nm), mCherry (570–700 nm), and mCarmine (650–770 nm) and specific fluorescence signals were acquired sequentially. Images were acquired and exported using the Leica Application Suite X software and further processed in Fiji software. Images were pseudo-colorized and brightness/contrast was adjusted. For co-localization analysis, middle slices of cells were imaged, exported, and analyzed using the Coloc2 plug-in of the Fiji software to yield Pearson's correlation coefficients (Pearson's r) of analyzed channels. Mean correlation coefficients are shown in Appendix Tables S6 and S7.

## Co-localization analysis in BMK/DKO cells

BMK/DKO cells expressing mTurquoise2-conjugated effector TMDs or mCerulean3-BIK were seeded in a 384-well plate (2000 cells/well). Twenty-four hours later, these cells were stained with Draq5, Mitotracker red, and BODIPY-Thapsigargin for 20 min at 37 °C followed by imaging. Images were taken using a confocal Opera Phenix High-Content Screening System with a 63× water-immersion Objective (Perkin Elmer/revvity, Waltham, MA, USA) and a total of 1000 cells was randomly chosen for subsequent analysis. Pearson's correlation coefficients between mTurquoise2/mCerulean3 and Mitotracker red or ER-marker BODIPY-Thapsigargin were determined in cytoplasmic regions of individual cells by excluding the Draq5-stained nuclei. Mean correlation coefficients are shown in Appendix Table S6.

## NanoBiT luminescence assay

Indicated combinations of SmBiT- and LgBiT-TMD fusion proteins were expressed in cells by transient transfection of plasmids in a 1:1 ratio and incubation for 24 h. Cells were harvested, washed in PBS and resuspended in 150 μl Opti-MEM Reduced Serum Medium (Thermo Fisher Scientific). Luminescence and fluorescence were assessed in technical triplicates (50 μl cell suspension/well) in a white F-bottom 96-well plate (Greiner Bio-One, Frickenhausen, Germany) using an EnSpire multimode plate reader (PerkinElmer, Waltham, MA, USA). To each well 50 μl luciferase substrate coelenterazine-h (Promega) in Opti-MEM was added to obtain a final concentration of 5 μM and luminescence as well as fluorescence were detected immediately afterwards. Initially, fluorescence was assessed once for mTurquoise2 (excitation at 434 nm/detection at 474 nm) and mCitrine (excitation at 516 nm/detection at 531 nm). Then, luminescence intensity was detected every 5 min for at least 30 min after substrate addition with a measurement duration of 1 s/well. For comparison of different samples, luminescence intensities 30 min after substrate addition were normalized to detected mTurquoise2 fluorescence. Normalized luminescence values for each technical triplicate were combined to a mean for each independent experiment.

## High-throughput multiscaling molecular dynamics simulations

Spontaneous self-association of BOK-TMD (comprising residues 180-S YNPGL RSHWL VAALC SFGRF LKAAF FVLLP ER-212)

and BCL-2-TMD (208-PLF DFSWL SLKTL LSLAL VGACI TLGAY LGHK-239) was studied by DAFT (Wassenaar et al, 2015) using the Martini3 (Souza et al, 2021) coarse-grained force-field in a membrane mimic of the ER membrane at 37 °C. The membrane model consisted of 75% 1-myristoyl-2-oleoyl-sn-glycero-3-phosphocholine (MOPC), 7% 1-myristoyl-2-oleoyl-sn-glycero-3-phosphoethanolamine (MOPE), 7% 1-palmitoyl-2-oleoyl-sn-glycero-3-phosphoinositol (POPI), 4% a-palmitoyl-2-oleoyl-sn-glycerol (PODG) and 7% cholesterol distributed symmetrically in the two membrane leaflets. After spontaneous association at CG resolution, the oligomers were clustered and representatives of the most often occurring conformations were converted back to atomistic resolution using backward (Wassenaar et al, 2014) and re-equilibrated atomistically for 1 μs using the CHARMM36m (Huang et al, 2017) force field and the TIP4p water model (Jorgensen and Madura, 1985). All simulations were performed and analyzed using GROMACS 2020 (Abraham et al, 2015) and visualized in PyMOL (Schrödinger, 2023). Gnuplot (Williams and Kelley, 2013) was used to generate contact maps and secondary structure plots. Overview of performed simulations is shown in Appendix Table S8. Details on simulation setup and conditions are provided in the Appendix.

## Statistical analysis

Statistical significance of differences was calculated by one-way ANOVA with Tukey's or Dunnett's multiple comparison test as indicated using GraphPad Prism 9 software. Significance is denoted as $*p < 0.05$, $**p < 0.01$, $***p < 0.001$, or $****p < 0.0001$.

# Data availability

This work does not include data deposited in a public database.

The source data of this paper are collected in the following database record: biostudies:S-SCDT-10_1038-S44319-024-00206-6.

# Peer review information

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

## Acknowledgements

Thanks go to Oliver Burk for kindly providing the NanoBiT system. Likewise, we thank Oliver Griesbeck and Arne Fabritius for kindly providing the mCarmine subcellular marker. MD simulations were performed on the HoreKa supercomputer funded by the Ministry of Science, Research and the Arts Baden-Württemberg and by the Federal Ministry of Education and Research. This work was supported by the Deutsche Forschungsgemeinschaft under Germany's Excellence Strategy – EXC 2075 – 390740016, the Stuttgart Center for Simulation Science (SC SimTech) and the Robert Bosch Stiftung GmbH, and by grant FDN143312 from the Canadian Institutes of Health Research.

## Author contributions

**Tobias B Beigl**: Conceptualization; Data curation; Formal analysis; Investigation; Visualization; Methodology; Writing—original draft; Writing—review and editing. **Alexander Paul**: Formal analysis; Investigation; Visualization; Writing—review and editing. **Thomas P Fellmeth**: Data curation; Formal analysis; Investigation; Visualization; Writing—review and editing. **Dang Nguyen**: Conceptualization; Data curation; Formal analysis; Validation; Investigation; Visualization; Methodology; Writing—review and editing. **Lynn Barber**: Formal analysis; Investigation; Writing—review and editing. **Sandra Weller**: Formal analysis; Investigation; Writing—review and editing. **Benjamin Schäfer**: Formal analysis; Investigation; Writing—review and editing. **Bernhard F Gillissen**: Data curation; Formal analysis; Supervision; Investigation; Writing—original draft; Writing—review and editing. **Walter E Aulitzky**: Funding acquisition; Project administration; Writing—review and editing. **Hans-Georg Kopp**: Conceptualization; Supervision; Funding acquisition; Project administration; Writing—review and editing. **Markus Rehm**: Conceptualization; Resources; Supervision; Validation; Methodology; Project administration; Writing—review and editing. **David W Andrews**: Conceptualization; Resources; Data curation; Formal analysis; Supervision; Funding acquisition; Validation; Investigation; Methodology; Project administration; Writing—review and editing. **Kristyna Pluhackova**: Conceptualization; Data curation; Software; Formal analysis; Supervision; Funding acquisition; Validation; Investigation; Visualization; Methodology; Writing—original draft; Project administration; Writing—review and editing. **Frank Essmann**: Conceptualization; Resources; Data curation; Formal analysis; Supervision; Funding acquisition; Validation; Investigation; Visualization; Methodology; Writing—original draft; Project administration; Writing—review and editing.

Source data underlying figure panels in this paper may have individual authorship assigned. Where available, figure panel/source data authorship is listed in the following database record: biostudies:S-SCDT-10_1038-S44319-024-00206-6.

## Disclosure and competing interests statement

The authors declare no competing interests.

# Expanded View Figures

**Figure EV1.   (belonging to Fig. 1): A bimolecular split luciferase assay reveals that TMDs of BCL-2 interact with TMDs of BOK but not with TMDs of BAX or BAK.** ▶

(**A**) Common structure of Bcl-2 family proteins. Shown are the BH (=Bcl-2 homology) motifs and TMD as boxes. Black dashed line indicates non-proportional representation. Position of alpha helices (α1-9) are indicated in dotted, light gray lines below. (**B**) Proportional co-expression of mCitrine and mTurquoise2. Scatter plots show mCitrine/mTurquoise2 fluorescence intensity detected in individual HEK293FT cells transfected with indicated LgBiT- and/or SmBiT-TMDs. Cells were analyzed 24 h after transfection using flow cytometry. Representative graphs out of three independent experiments. (**C**) Titration of HEK293FT cells transfected with increasing amounts of plasmid for the expression of SmBiT-BAX-TMD, while keeping co-transfected LgBiT-BAX-TMD-encoding plasmid constant. To achieve equal total amount of DNA per transfection, SmBiT-BAX-TMD was mixed with empty backbone vector. Cells were harvested 24 h post transfection and subjected to both split luciferase assay as well as Western blot. Luminescence intensities are shown as mean ± sd of technical triplicates from one representative experiment relative to transfection without SmBiT-BAX-TMD-encoding plasmid. Detection of LgBiT and mTurquoise2 (mTurq2) expression in same samples using Western blot is shown below. β-actin was used as loading control. (**D**) Luminescence of HEK293FT cells co-transfected with plasmids for the expression of LgBiT-BAX-TMD and SmBiT-TOM5-TMD or SmBiT-BAX-TMD, harvested after 24 h and subjected to the split luciferase assay. Detected luminescence was normalized to simultaneously acquired mTurquoise2 fluorescence (mTurq2) or mCitrine fluorescence (mCitr). Mean ± sd from three independent experiments shown in relation to simultaneously detected positive control (BAX-TMD/BAX-TMD, set to 1.00). (**E**) Split luciferase assay confirming BOK-TMD/BCL-2-TMD interaction in MCF-7 cells. MCF-7 cells were transfected with plasmids for the expression of indicated NanoBiT-TMD fusion proteins and harvested after 24 h to be subjected to NanoBiT assay. Luminescence intensities shown as mean ± sd from three independent experiment were set in relation to positive control (BAX-TMD/BAX-TMD). ****$p < 0.0001$, one-way ANOVA with Tukey's multiple comparison test.

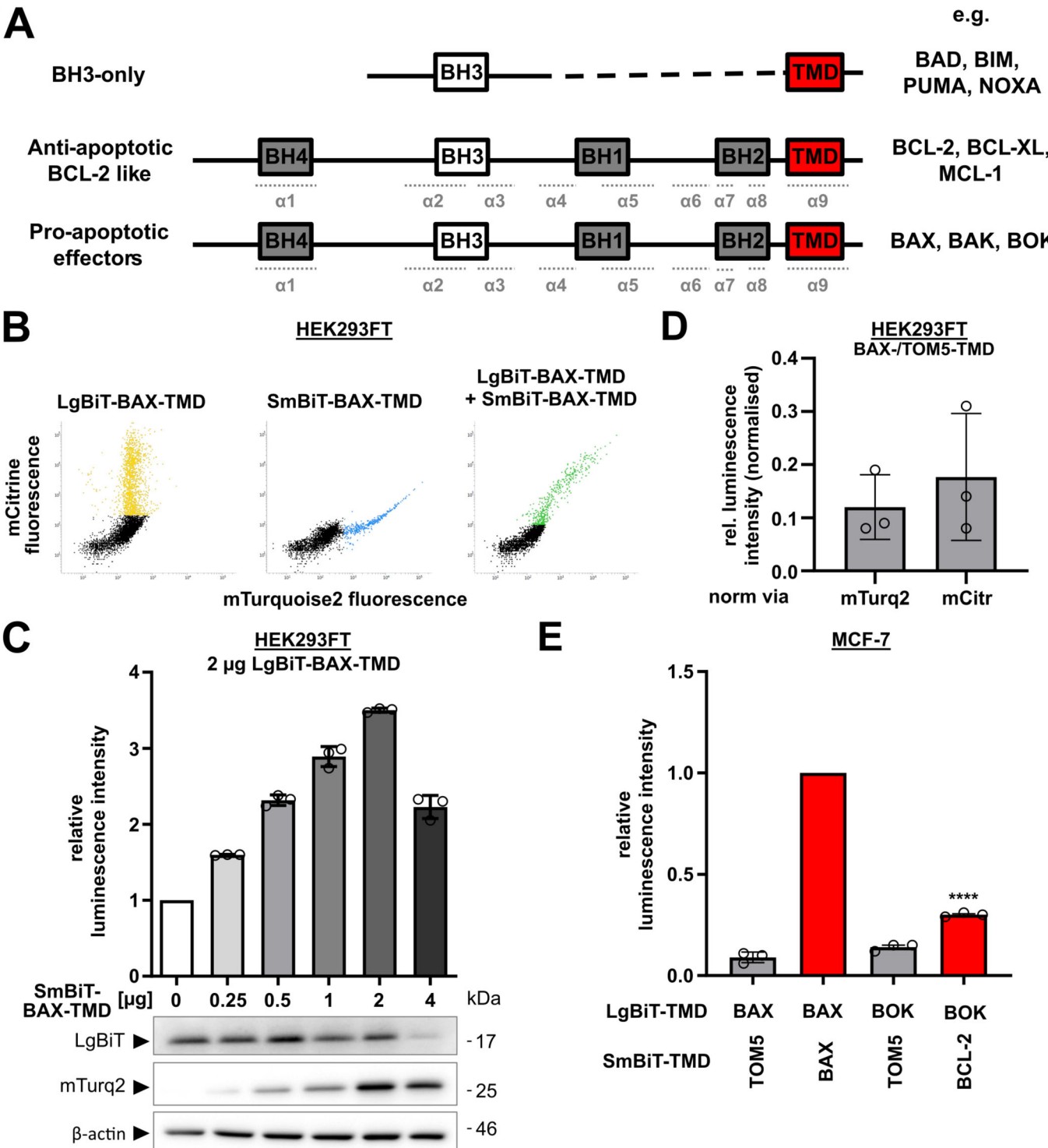

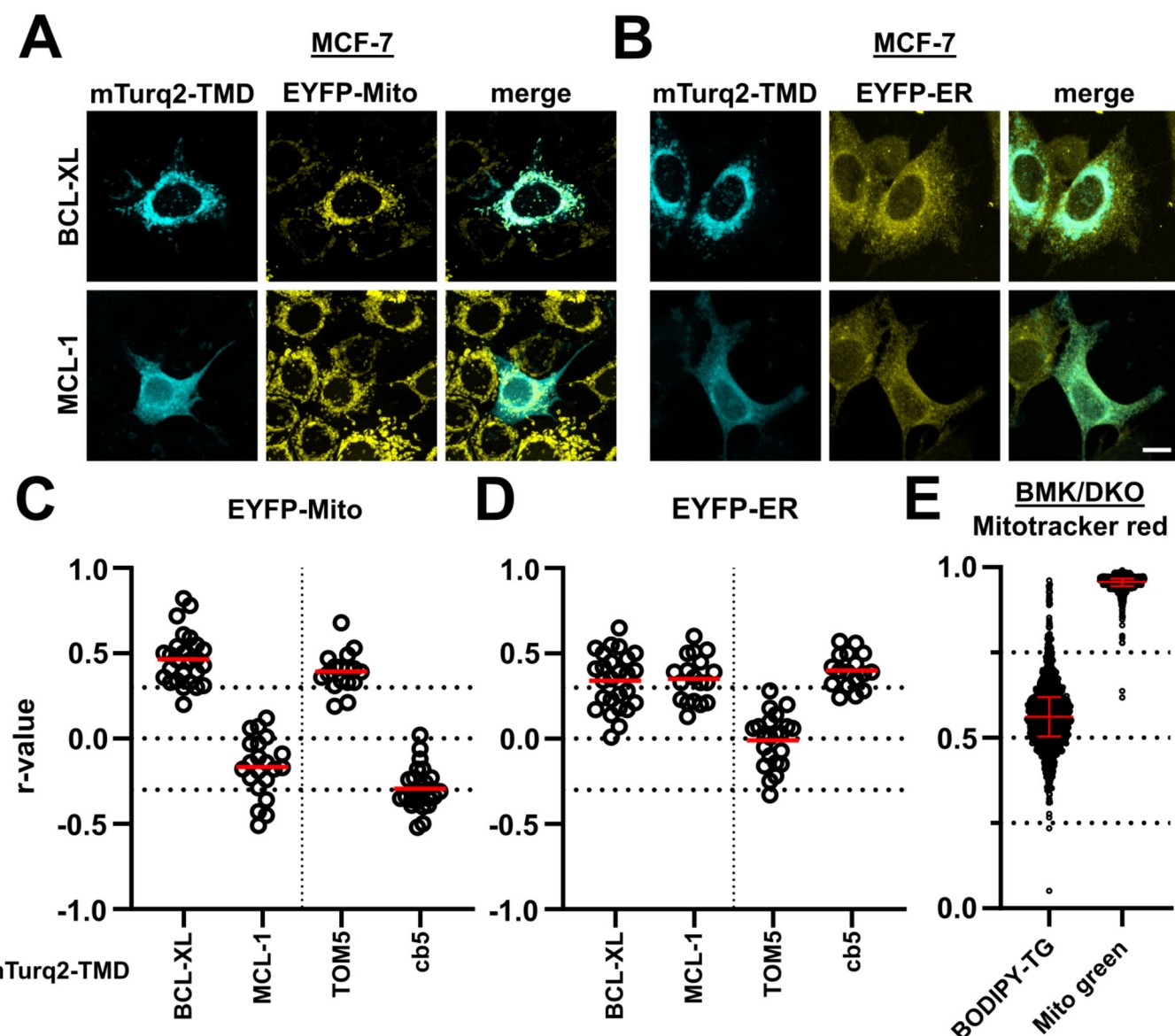

**Figure EV2.  (belonging to Fig. 2): BCL-2-TMD and BOK TMD are predominantly ER-localized.**

(A, B) Subcellular localization of MCL-1-TMD and BCL-XL-TMD. MCF-7 cells expressing (A) EYFP-Mito or (B) EYFP-ER were transfected with plasmids for the expression of mTurquoise2 (mTurq2)-labeled BCL-XL-TMD or MCL-1-TMD and 24 h post transfection cells were imaged by cLSM. Images are maximum projections of representative z-stacks from three independent experiments. Scale bar = 10 μm. (C, D) Quantitative analysis of mTurquoise2-fused BCL-XL-TMD or MCL-1-TMD co-localization with (C) EYFP-labeled mitochondria and (D) EYFP-labeled ER from cLSM images (A, B). Graphs show Pearson's r correlation coefficient for a total of ≥15 cells combined from three independent experiments. The mean is marked as a red horizontal line. Data for mTurquoise-TOM5-TMD and mTurquoise2-cb5-TMD are shown for comparison. (E) Co-localization analysis of subcellular markers in BMK/DKO cells. BMK/DKO cells labeled with DRAQ5 (nuclei) and Mitotracker red were stained with BODIPY-Thapsigargin (BODIPY-TG, ER control) or Mitotracker green (Mito control). Images of n ≥ 1000 cells were acquired using an Opera Phenix spinning disk system, images were analyzed and Pearson's r calculated. Shown in red are the median + IQR.

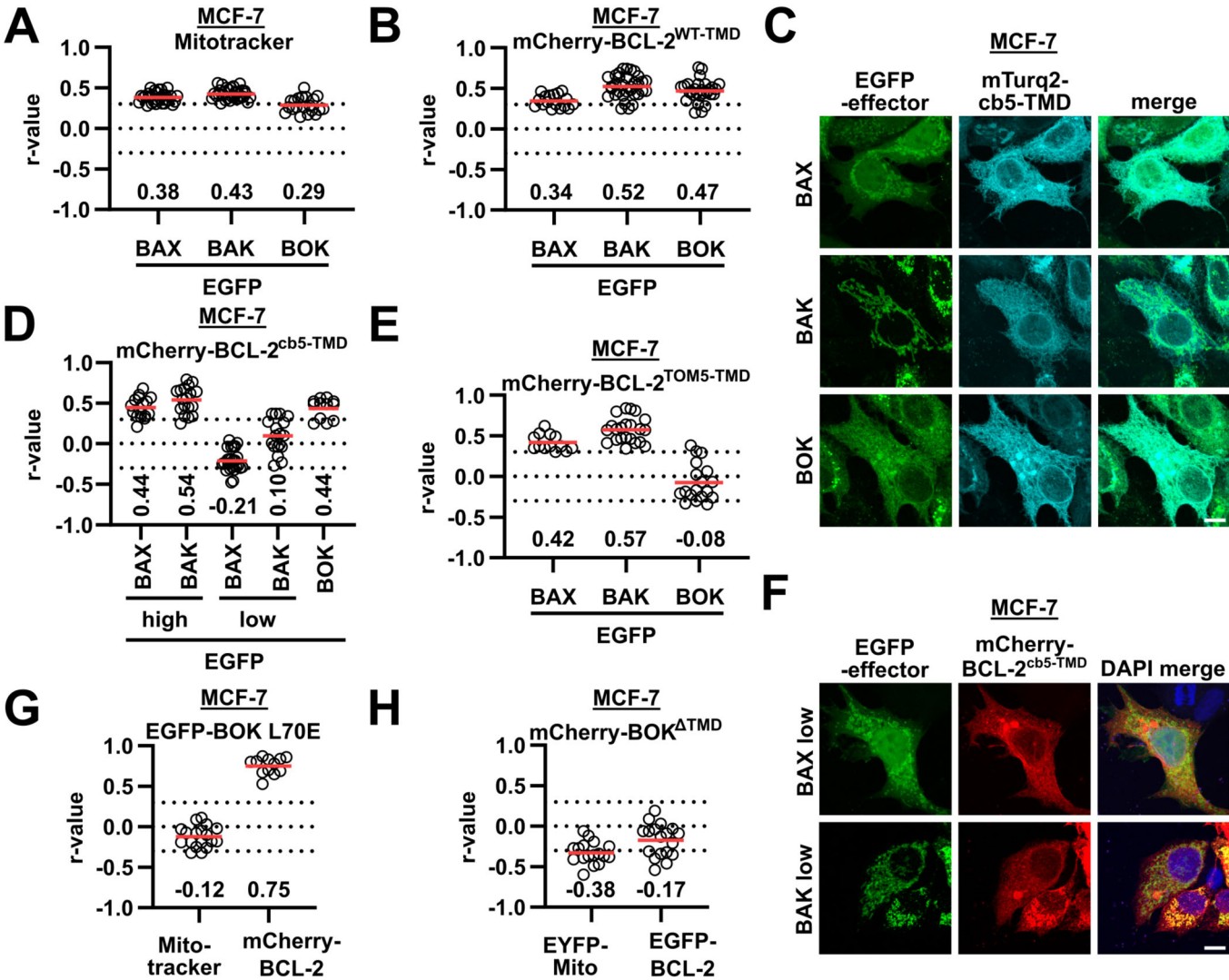

**Figure EV3.** (belonging to Fig. 4): BOK and BCL-2 co-localization at the ER is dictated by their TMDs.

(A, B) Quantitative analysis of co-localization from middle sections of cLSM images as shown in Fig. 4C,D. Scatter plots show Pearson's r correlation coefficients of in total ≥10 single cells from two independent experiments determined using Fiji software. Mean is marked as a red horizontal line. Co-localization between EGFP-effectors (BAX, BAK, and BOK) and Mitotracker (A) or mCherry-BCL-2 (B). (C) Co-localization of BAX, BAK, and BOK with ER (cb5-TMD). MCF-7 cells were co-transfected with plasmids for the expression of EGFP-BAX, EGFP-BAK or EGFP-BOK and mTurq2-cb5-TMD. cLSM images are maximum projections of z-stacks from two independent experiments. Scale bar = 10 μm. (D, E) Quantitative analysis of co-localization as in (A, B) from Fig. 4E,F and Fig. EV3F. Co-localization between EGFP-effectors (BAX, BAK, and BOK) and mCherry-BCL-2cb5-TMD (D) or mCherry-BCL-2TOM5-TMD (E). In (D), Co-localization in cells with high/low BAX/BAK expression was analyzed separately. (F) Co-localization of BAX and BAK (low expression) with BCL-2cb5-TMD. MCF-7 cells were co-transfected with plasmids for the expression of EGFP-BAX or EGFP-BAK and mCherry-BCL-2cb5-TMD. Shown images represent cells without cluster formation of BAX or BAK. cLSM images are maximum projections of z-stacks from two independent experiments. Scale bar = 10 μm. (G, H) Quantitative analysis of co-localization as in (A, B) from Fig. 4G,H. (G) Co-localization between EGFP-BOK L70E and Mitotracker red or mCherry-BCL-2. (H) Co-localization between mCherry-BOKΔTMD and EYFP-Mito or EGFP-BCL-2.

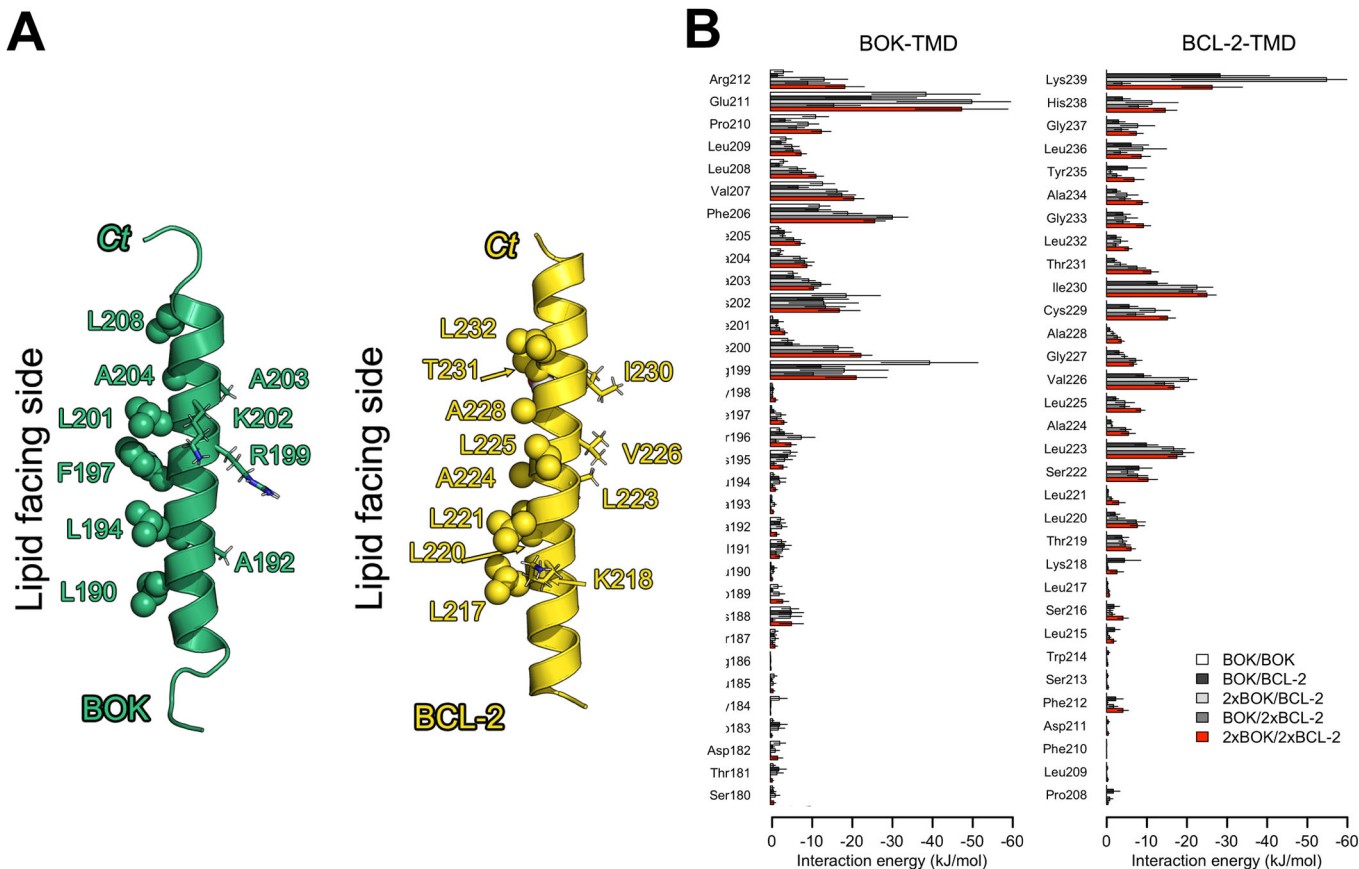

**Figure EV4.** (belonging to Fig. 6): Interaction of BOK-TMD and BCL-2-TMD depends on specific residues in the interaction interface.

(A) Visualization of BCL-2-TMD and BOK-TMD helices with membrane anchors (K218$^{BCL-2}$ and R199$^{BOK}$ and K202$^{BOK}$) and interaction interface residues (I230$^{BCL-2}$ V226$^{BCL-2}$ L223$^{BCL-2}$, A203$^{BOK}$ A192$^{BOK}$). The lipid facing residues are highlighted as spheres. (B) Average interaction energies of each residue of BOK-TMD (left) or BCL-2-TMD (right) with all other peptides over the last 100 ns of AA simulations of dimers, trimer, and tetramers. The error bars denote SEM over individual simulations. The outer error bars of E211$^{BOK}$ in 2xBOK/BCL-2 trimer and K239$^{BCL-2}$ 2xBOK/BCL-2 trimer simulations are cut at −60 kJ/mol due to visualization purposes.

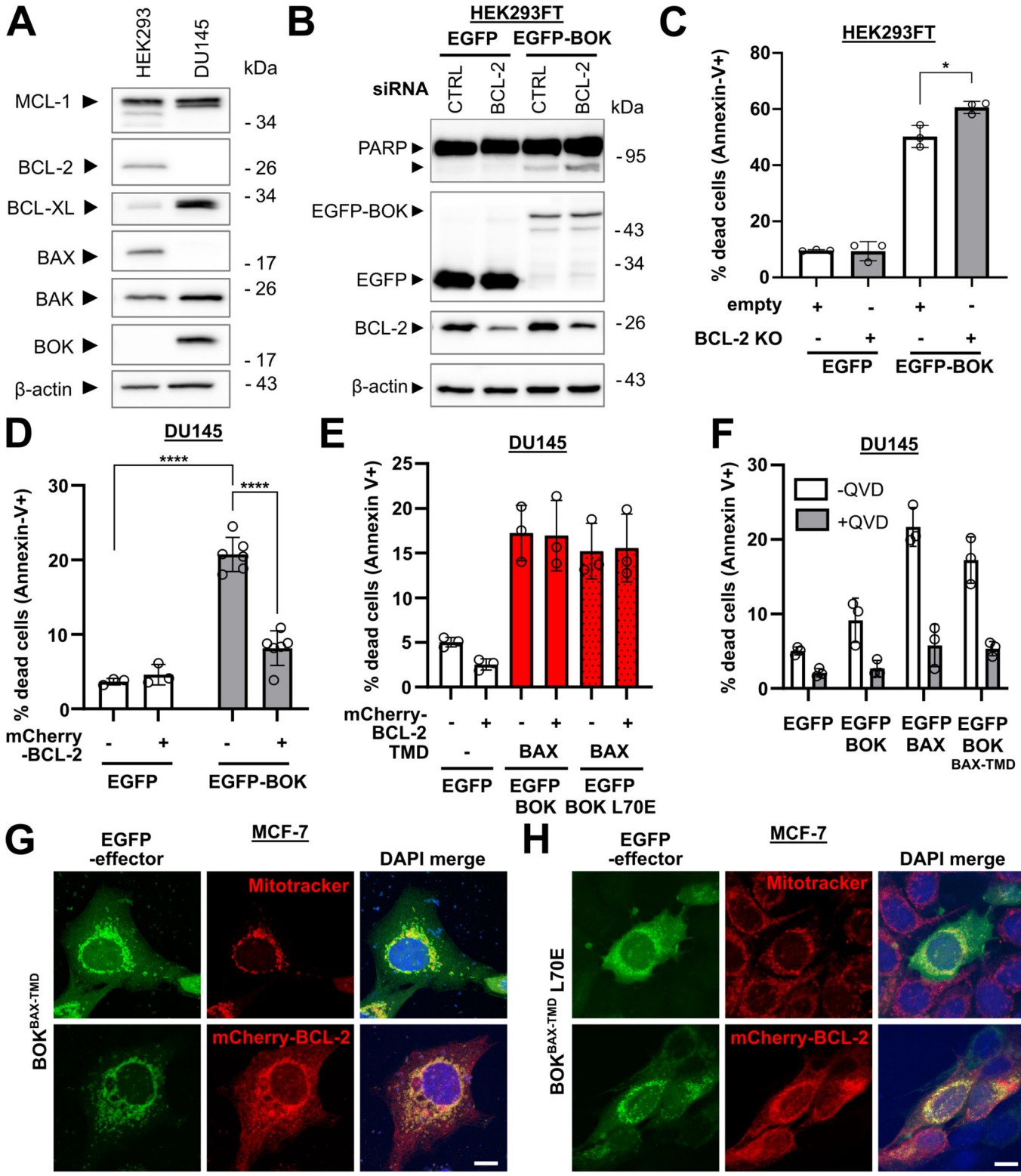

◀ **Figure EV5.   (belonging to Fig. 7): Inhibition of BOK-mediated apoptosis by BCL-2 depends on BOK-TMD.**

(**A**) Detection of various Bcl-2 family proteins in whole cell lysates of HEK293FT and DU145 cells using Western blot. β-actin was used as loading control. One independent experiment. (**B**) HEK293FT cells were transfected with siRNA for BCL-2 (siBCL-2) or control siRNA (siCTRL) for 24 h and were subsequently transfected with plasmids for the expression of EGFP-BOK or EGFP. After 18 h, cells were harvested and analyzed by Western blot. PARP, EGFP(-BOK) and BCL-2 expression were detected. β-actin was used as loading control. Representative blot from two independent experiments. (**C**) HEK293FT cells were co-transfected with plasmids for the expression of a CRISPR/Cas9 vector targeting *BCL2* and EGFP or EGFP-BOK (empty = empty vector backbone control). After 18 h, cell death was assessed using Annexin-V-APC staining and flow cytometry. EGFP+/Annexin-V+ cells were detected by flow cytometry. Mean ± sd from three independent experiments. *$p < 0.05$, one-way ANOVA with Tukey's multiple comparison test. (**D**) DU145 cells were transfected with plasmids for the expression of EGFP-BOK or EGFP in combination with mCherry-BCL-2 or an empty vector as a control. After 42 h, cells were stained with Annexin-V-APC and cell death (EGFP+/Annexin-V+ cells) was assessed using flow cytometry. Mean ± sd from three (EGFP) or six (EGFP-BOK) independent experiments. ****$p < 0.0001$, one-way ANOVA with Tukey's multiple comparison test. (**E**) DU145 cells were transfected with plasmids for the expression of EGFP, EGFP-BOK^BAX-TMD, or EGFP-BOK^BAX-TMD L70E in combination with mCherry-BCL-2 or an empty vector as a control. After 18 h, cells were stained with Annexin-V-APC and cell death (EGFP+/Annexin-V+ cells) was assessed using flow cytometry. Mean ± sd from three independent experiments. (**F**) DU145 cells were transfected with plasmids for the expression of EGFP, EGFP-BOK, EGFP-BAX and EGFP-BOK^BAX-TMD and incubated with 10 μM pan-caspase inhibitor QVD-OPh (+QVD) or DMSO as control (-QVD). After 18 h, cells were stained with Annexin-V-APC and cell death (EGFP+/Annexin-V+ cells) was assessed using flow cytometry. Mean ± sd from three independent experiments. (**G, H**) MCF-7 cells were transfected with plasmids for the expression of either EGFP-BOK^BAX-TMD (**G**) or EGFP-BOK^BAX-TMD L70E (**H**). Cells were either stained with Mitotracker red after 24 h or co-transfected to express mCherry-BCL-2. Twenty-four hours post transfection cells were fixed and analyzed by cLSM. Images are maximum projection of z-stacks representative of two independent experiments. Scale bar = 10 μm.

                                                            