## [Peer Review File · EMBO Reports]

BCL-2 and BOK regulate apoptosis by interaction of their C-terminal transmembrane domains

Tobias Beigl, Alexander Paul, Thomas Fellmeth, Dang Nguyen, Lynn Barber, Sandra Weller, Benjamin Schaefer, Bernhard Gillissen, Walter Aulitzky, Hans-Georg Kopp, Markus Rehm, David Andrews, Kristyna Pluhackova, and Frank Essmann

Corresponding author(s): Frank Essmann (frank.essmann@bosch-health-campus.com)

Review Timeline:

Submission Date:	7th Sep 23
Editorial Decision:	9th Oct 23
Revision Received:	6th Feb 24
Editorial Decision:	20th Mar 24
Revision Received:	17th Jun 24
Accepted:	25th Jun 24

Transaction Report:

Dear Dr. Essmann

Thank you for the submission of your research manuscript to our journal. We have now received the full set of referee reports that is copied below.

As you will see, the referees acknowledge that the findings are potentially interesting, but they also raise a number of concerns, which need to be addressed.

Given their constructive comments, we would like to invite you to revise your manuscript with the understanding that the referee concerns (as detailed above and in their reports) must be fully addressed and their suggestions taken on board. Please address all referee concerns in a complete point-by-point response. Acceptance of the manuscript will depend on a positive outcome of a second round of review. It is EMBO Reports policy to allow a single round of revision only and acceptance or rejection of the manuscript will therefore depend on the completeness of your responses included in the next, final version of the manuscript.

We realize that it is difficult to revise to a specific deadline. In the interest of protecting the conceptual advance provided by the work, we recommend a revision within 3 months (January 9th, 2024). Please discuss the revision progress ahead of this time with the editor if you require more time to complete the revisions.

I am also happy to discuss the revision further via e-mail or a video call, if you wish.

*****IMPORTANT NOTE:

We perform an initial quality control of all revised manuscripts before re-review. Your manuscript will FAIL this control and the handling will be delayed IN CASE the following APPLIES:

- 1) A data availability section providing access to data deposited in public databases is missing. If you have not deposited any data, please add a sentence to the data availability section that explains that.
- 2) Your manuscript contains statistics and error bars based on $n=2$. Please use scatter blots in these cases. No statistics should be calculated if $n=2$.

When submitting your revised manuscript, please carefully review the instructions that follow below. Failure to include requested items will delay the evaluation of your revision.*****

- 1) a .docx formatted version of the manuscript text (including legends for main figures, EV figures and tables). Please make sure that the changes are highlighted to be clearly visible.
- 2) individual production quality figure files as .eps, .tif, .jpg (one file per figure). Please download our Figure Preparation Guidelines (figure preparation pdf) from our Author Guidelines pages <https://www.embopress.org/page/journal/14693178/authorguide> for more info on how to prepare your figures.
- 3) a .docx formatted letter INCLUDING the reviewers' reports and your detailed point-by-point responses to their comments. As part of the EMBO Press transparent editorial process, the point-by-point response is part of the Review Process File (RPF), which will be published alongside your paper.
- 4) a complete author checklist, which you can download from our author guidelines (<<https://www.embopress.org/page/journal/14693178/authorguide>>). Please insert information in the checklist that is also reflected in the manuscript. The completed author checklist will also be part of the RPF.
- 5) Please note that all corresponding authors are required to supply an ORCID ID for their name upon submission of a revised manuscript (<<https://orcid.org/>>). Please find instructions on how to link your ORCID ID to your account in our manuscript tracking system in our Author guidelines (<<https://www.embopress.org/page/journal/14693178/authorguide#authorshipguidelines>>)
- 6) We replaced Supplementary Information with Expanded View (EV) Figures and Tables that are collapsible/expandable online. A maximum of 5 EV Figures can be typeset. EV Figures should be cited as "Figure EV1, Figure EV2" etc... in the text and their respective legends should be included in the main text after the legends of regular figures.

7) Please note that a Data Availability section at the end of Materials and Methods is now mandatory. In case you have no data that requires deposition in a public database, please state so instead of refereeing to the database. See also <<https://www.embopress.org/page/journal/14693178/authorguide#dataavailability>>. Please note that the Data Availability Section is restricted to new primary data that are part of this study.

Additional information on source data and instruction on how to label the files are available <<https://www.embopress.org/page/journal/14693178/authorguide#sourcedata>>.

10) Figure legends and data quantification:
The following points must be specified in each figure legend:

- the name of the statistical test used to generate error bars and P values,
 - the number (n) of independent experiments (please specify technical or biological replicates) underlying each data point,
 - the nature of the bars and error bars (s.d., s.e.m.)
- If the data are obtained from n {less than or equal to} 5, show the individual data points in addition to the SD or SEM.
 - If the data are obtained from n {less than or equal to} 2, use scatter blots showing the individual data points.

See also the guidelines for figure legend preparation:
<https://www.embopress.org/page/journal/14693178/authorguide#figureformat>

11) Our journal encourages inclusion of *data citations in the reference list* to directly cite datasets that were re-used and obtained from public databases. Data citations in the article text are distinct from normal bibliographical citations and should directly link to the database records from which the data can be accessed. In the main text, data citations are formatted as follows: "Data ref: Smith et al, 2001" or "Data ref: NCBI Sequence Read Archive PRJNA342805, 2017". In the Reference list, data citations must be labeled with "[DATASET]". A data reference must provide the database name, accession number/identifiers and a resolvable link to the landing page from which the data can be accessed at the end of the reference. Further instructions are available at <<https://www.embopress.org/page/journal/14693178/authorguide#referencesformat>>.

12) All Materials and Methods need to be described in the main text. We would encourage you to use 'Structured Methods', our new Materials and Methods format. According to this format, the Materials and Methods section should include a Reagents and Tools Table (listing key reagents, experimental models, software and relevant equipment and including their sources and relevant identifiers) followed by a Methods and Protocols section in which we encourage the authors to describe their methods using a step-by-step protocol format with bullet points, to facilitate the adoption of the methodologies across labs.

More information on how to adhere to this format as well as downloadable templates (.doc or .xls) for the Reagents and Tools Table can be found in our author guidelines: <<https://www.embopress.org/page/journal/14693178/authorguide#manuscriptpreparation>>.
An example of a Method paper with Structured Methods can be found here: <<https://www.embopress.org/doi/10.15252/msb.20178071>>.

13) As part of the EMBO publication's Transparent Editorial Process, EMBO Reports publishes online a Review Process File to accompany accepted manuscripts. This File will be published in conjunction with your paper and will include the referee reports, your point-by-point response and all pertinent correspondence relating to the manuscript.

Yours sincerely,

Referee #1:

In this paper, Beigl et al. show for the first time an interaction between the pro-apoptotic BCL-2 family member BOK and anti-apoptotic BCL-2. By using a bimolecular split luciferase assay and the expression of fluorescently tagged proteins, they find that the interaction between BOK and BCL-2 is not via the classical BH3- but via their transmembrane domains (TMD). The interaction occurs on the ER where BOK and BCL-2 were reported to mainly localize, and it has functional consequence as the inhibition of BOK-induced apoptosis by BCL-2 is dependent on their interaction their TMDs.

The data have been obtained by many different methods, show interactions between the BCL-2 family members and their TMDs in vitro and inside cells, include functional analyses and are well controlled and statistically significant. I think this study deserves publication in EMBO Reports after some points have been addressed:

Major points:

1) The choice of the fluorophores mCitrine and mTurquoise2 for interaction studies are sometimes a bit unfortunate. The blue and yellow colors provide a greenish colocalization stain which is difficult to see and interpret. One can nicely see if co-localization does NOT occur, in which case there are still yellow spots in the double stain (For example Figure 2A, BOK, BCL-2, cb5). However, it is not easy to detect co-localization on the ER with the ER-marker EYFP-ER (Figure 2B). This is also true for the data shown in Figures S2A and B. Here Mcl-1 unexpectedly does not seem to localize to mitochondria (yellow spots in Figure S2A). However, it is hard to see that Mcl-1 indeed localizes to the ER instead (Figure S2B).

The calculations of the r-values (Pearson's correlation) of the pictures shown in Figures 2 and S2 seem to work and be significant (Figures 2C-E, Figures S2C-E) but I wonder how the authors could obtain these clear results. The co-localization on the ER (and on mitochondria) can be much better seen with the combination EGFP and mCherry (as it is shown in Figure 4). I suggest to redo some of the TMD interactions studies shown in Figures 2 and S2 with the EGFP/mCherry combination.

2) While cell death induction by WT EGFP-BOK is inhibited by mCherry-BCL-2 co-expression, cell death induced by EGFP-BOK;BAX-TMD was not altered by co-expression of mCherry-BCL-2 (Figure 6B). The authors interpret this by stating that BCL-2 cannot inhibit BOK localized on mitochondria (which seems to be the case based on Figure S3C). However, two control experiments are missing. 1) In Figure S3C it is not really shown that upon co-expression with EGFP-BOK;BAX-TMD, mCherry BCL-2 remains ER-bound. The co-staining picture is not clear. 2) Since BAX recruits BCL-2 to mitochondria a BH3 and not a TMD interaction, this does not seem to be the case with EGFP-BOK;BAX-TMD. To prove this, the authors should express the a BH3-L70E mutant of EGFP-BOK;BAX-TMD to show that it also does not attract mCherry-BCL-2 to mitochondria. This mutant would also be interesting for functional assays in Figure 6 to show if EGFP-BOK-mediated cell death is indeed BH3-dependent (both on the ER and on mitochondria). If cell death is independent of BH3, then it should still be inhibited by mCherry-BCL-2 via TMD interaction.

3) While EGFP-BOK;BAX-TMD expression led to more Annexin-V+ apoptotic DU145 cells than EGFP-BOK expression (Figure 6B) due to a higher stability of mitochondrially localized EGFP-BOK;BAX-TMD, this difference is not seen when apoptosis was measured by caspase-3 activity (Figure 6F). Why is there a discrepancy? Does this mean that EGFP-BOK;BAX-TMD majorly induces caspase-independent cell death when targeted to mitochondria? This is an interesting finding that needs to be discussed.

Minor points:

Lines 235-236: The MCL-1-TMD data are not shown in Figure S2A. This should be indicated.

Lines 282-293: The co-localization studies are shown in Figure 4C and Figure S3A and not S4A).

Referee #2:

This manuscript provides a very detailed interrogation of the binding partners of the transmembrane domain (TMD) of the Bcl2 family member BOK and provides a structural basis for why this protein behaves differently from other members of the family. For the most part experiments are well presented and controlled with suitable quantitation. The major weakness is in Figure 6 where, as detailed below, the effects on apoptosis are just too small to provide meaningful conclusions on the role or necessity of BOK for apoptosis.

Major concerns.

1. There are 19 supplemental figures. Some of them should be included in the main figures. For example S1B would be better than the current 1C as the images only show that the linked flours are expressed. Another example is Figure S2F. If others could be consolidated into fewer figures it may make it a little easier to follow.
2. Most of the imaging comes with very nice quantitation except for Figure 4. It seems that the co-localization coefficients are only representing 1 image. Quantitation similar to the other figures should be included. Also, it is stated on line 281 that analysis was only done on cells without clustered EGFP signals. What proportion of cells does this represent? Similarly how were the high vs low levels of Bax determined?
3. I am not a structural biologist so I cannot fully critique the methods and interpretations of the molecular docking data. It would have been nice to test some mutants of the identified residues to validate the model but I agree that this may be outside the scope of the current manuscript.
4. Figure 6 uses mostly overexpression to assess BOK induced apoptosis. The problem is that the amount of death induced by BOK is not very good ranging from 10-20% in DU145 cells or not very well inhibited by Bcl2 at around 10%. Although these differences are statistically significant, they are likely not biologically relevant especially when Bcl2 expression usually induces full resistance in other contexts. Figure 6C is particularly problematic because instead of showing the raw data the corrected percentage inhibition is shown. This could equally be 100% cell death going down to 5% or 5% going down to 1 which is not as effective. Due to the small effects, one could conclude that there are issues with the overexpression based system not fully recapitulating the functions of these proteins OR that the BOK and BCL2 interaction regulates something other than cell death. It is difficult to fully conclude either based on the data as shown. Investigating the Bok dependence of cell death induced by different stimuli using CRISPR or RNAi with or without Bcl2 may be a better approach.

Minor

1. A number of the supplemental figures are incorrectly labeled in the text
2. In Figure 2D, it would be helpful to include a gate on what is considered double positive cells in the flow plots
3. Some of the confocal images appear a little oversaturated especially the EGFP and TFP images. Suggest dialing them back a little
4. What is the rationale for using the Bax/Bak DKO cells?
5. The intro is a little too long and the whole manuscript could use a good proofread, otherwise it is nicely written

Referee #3:

Since Bcl-2 was discovered as an oncogene inhibiting cancer cell apoptosis in the 80s, and cytochrome c was determined as an activator of caspases and was released from mitochondria by Bax in the 90s, mitochondrial dependent apoptosis regulated by Bcl-2 and other family proteins has been the focus for decades. As a result, the molecular mechanism by which Bcl-2s act to control the protein permeability of the mitochondrial membrane has been largely established. Structure and function-based translational research has resulted in a small molecule Bcl-2 inhibitor that has been used to treat blood cancer patients. In contrast, although Bcl-2s were known early on not only localizing to the mitochondria but other intracellular membranes, research of Bcl-2s structure and function in the membranes other than the mitochondrial ones has not been proportional. In particular the interactions within the Bcl-2 family were poorly defined, and the consequences of the interactions were mostly unknown.

In the manuscript submitted to EMBO Report, Beigl et al. concluded that Bcl-2 and Bok regulate apoptosis by interaction of their C-terminal transmembrane domains (TMDs) embedded in the ER membrane. The conclusion was based on the following evidence.

1. The interaction between the TMDs (not as parts of the full-length proteins) occurred in cells as revealed by a novel

biomolecular split luciferase assay.

2. The TMDs were colocalized at the ER in cells as imaged by super-resolution fluorescence microscopy.
3. The TMDs in the full-length proteins were critical for their colocalization at the ER because the colocalization was reduced when the TMD of Bcl-2 was replaced by the TMD of TOM5 that localized Bcl-2 to the mitochondria.
4. The BH3 domain of Bok and the BH3-binding groove of Bcl-2 was not critical to the colocalization because if this binding interaction still occurred between the Bcl-2 mutant with the TOM5 TMD, the Bok at the ER would attract the Bcl-2 mutant to the ER, or vice versa. A corroborating evidence is that mutation of a conserved L70 residue in the BH3 domain to E that would reduce the BH3-groove interaction did not affect the colocalization of Bok and Bcl-2, and that deletion of the TMD of Bok resulted in a cytosolic localization of Bok in the presence of Bcl-2 that are in the ER and mitochondria.
5. Structural models for BOK-Bcl-2 TMD interactions in an ER mimic lipid bilayer were generated by molecular dynamics simulations. These models suggest that the TMDs can not only form hetero dimers and oligomers but homo dimers and oligomers.
6. Distribution of certain ER lipids in the proximity of the TMD dimers and oligomers is different than the bulk lipid bilayer suggesting their preferential interactions with the TMD complexes.
7. The TMDs and their colocalization to the ER are important to Bcl-2 inhibition of Bok-induced apoptosis as determined by combinations of Bcl-2 and Bok mutants with different TMDs.

Thus, the overall conclusion is mostly based on experimental evidence. Moreover, molecular dynamics simulations provide structural models of molecular to atomic resolutions for the TMD complexes in the ER lipid bilayer.

Therefore, the manuscript would be suitable for publication. However, the following concern needs to be addressed to further solidify the conclusion and increase the impact of the work.

As part of the overall conclusion, the interaction of Bcl-2 and Bok that is mediated by their TMDs in the ER membrane is critical to the Bcl-2 inhibition of Bok-induced apoptosis. The work is then one step short from a full demonstration of the critical role of "the interaction between the TMDs in the ER membrane". What the authors might do is using the structural models of the Bcl-2-Bok TMD complexes generated by molecular dynamics simulations to identify point mutations in the complex interfaces, and then determine if the mutations reduce the complex formation in the ER membrane and the Bcl-2 inhibition of Bok in cells.

In addition, the following questions need to be answered during revision of the manuscript.

- A. Is there any structural model for how Bok and Bcl-2 interacts via the BH3-groove interface? If so, would the L70E mutation disrupt the complex? If not, could a such model be generate using AlphaFold2 or other AI structural biology tools? If still not, perhaps adding ABT-199 to the cells where Bcl-2 can inhibit Bok is a better experiment to do to see if the BH3-groove interaction mediates the Bok/Bcl-2 interaction and function.
- B. In the experiments for Figure 2, was any Bcl-2-TMD localize to the mitochondria? If not, please explain the difference between the TMD in the full-length Bcl-2 protein and the TMD by itself on the subcellular localization.
- C. There are many colocalizations compared throughout the text in Results using correlation coefficients. It would be better if they can be tabalized.
- D. What would be the effect of endogenous Bok and Bcl-2 on the experiments performed with different cells?
- E. Line 271-273, add another reference for Bak/Bcl-2 interaction via the BH3/groove binding interface.
- F. Some of the ER lipids would be enriched or depleted around the Bcl-2/Bok TMD complexes according to molecular dynamics simulations. Could this prediction be tested by experiments?
- G. Adding a schematic overall model for how Bcl-2 interacts with Bok at the ER to regulate Bok-induced apoptosis would be better for enhancing the visibility of this new discovery.
- H. The following papers could be referenced in the manuscript as early computational modeling and molecular dynamics simulations of Bax TMD homodimers in the mitochondrial lipid bilayer.

1) Zhang Z, Subramaniam S, Kale J, Liao C, Huang B, Brahmabhatt H, Condon SG, Lapolla SM, Hays FA, Ding J, He F, Zhang XC, Li J, Senes A, Andrews DW, Lin J. BH3-in-groove dimerization initiates and helix 9 dimerization expands Bax pore assembly in membranes. *EMBO J.* 2016 Jan 18;35(2):208-36. doi: 10.15252/embj.201591552. Epub 2015 Dec 23. PMID: 26702098; PMCID: PMC4718459.

2) Liao C, Zhang Z, Kale J, Andrews DW, Lin J, Li J. Conformational Heterogeneity of Bax Helix 9 Dimer for Apoptotic Pore Formation. *Sci Rep.* 2016 Jul 6;6:29502. doi: 10.1038/srep29502. PMID: 27381287; PMCID: PMC4933972.

Dear Senior Editor Martina Rembold,

Please find on the following pages our point-to-point reply to the reviewer's comments.

POINT-TO-POINT reply to reviewers' comments

Referee #1:

In this paper, Beigl et al. show for the first time an interaction between the pro-apoptotic BCL-2 family member BOK and anti-apoptotic BCL-2. By using a bimolecular split luciferase assay and the expression of fluorescently tagged proteins, they find that the interaction between BOK and BCL-2 is not via the classical BH3- but via their transmembrane domains (TMD). The interaction occurs on the ER where BOK and BCL-2 were reported to mainly localize, and it has functional consequence as the inhibition of BOK-induced apoptosis by BCL-2 is dependent on their interaction their TMDs.

The data have been obtained by many different methods, show interactions between the BCL-2 family members and their TMDs in vitro and inside cells, include functional analyses and are well controlled and statistically significant. I think this study deserves publication in EMBO Reports after some points have been addressed:

We thank the reviewer for the professional and positive evaluation and are grateful for the appreciation of our work.

Major points:

1) The choice of the fluorophores mCitrine and mTurquoise2 for interaction studies are sometimes a bit unfortunate. The blue and yellow colors provide a greenish colocalization stain which is difficult to see and interpret. One can nicely see if co-localization does NOT occur, in which case there are still yellow spots in the double stain (For example Figure 2A, BOK, BCL-2, cb5). However, it is not easy to detect co-localization on the ER with the ER-marker EYFP-ER (Figure 2B). This is also true for the data shown in Figures S2A and B. Here Mcl-1 unexpectedly does not seem to localize to mitochondria (yellow spots in Figure S2A). However, it is hard to see that Mcl-1 indeed localizes to the ER instead (Figure S2B). The calculations of the r-values (Pearson's correlation) of the pictures shown in Figures 2 and S2 seem to work and be significant (Figures 2C-E, Figures S2C-E) but I wonder how the authors could obtain these clear results. The co-localization on the ER (and on mitochondria) can be much better seen with the combination EGFP and mCherry (as it is shown in Figure 4). I suggest to redo some of the TMD interactions studies shown in Figures 2 and S2 with the EGFP/mCherry combination.

We thank the reviewer for this constructive recommendation. We agree that green/red is better suited to visually identify co-localization as yellow. We used a cLSM (Leica TCS SP8; 405 nm, 488 nm, 552 nm, 638 nm) for detection of fluorescence that allows to adjust the bandwidth to detect light with specific wavelengths using a photomultiplier and the images generally are pseudo-colored. Therefore, we do not see the advantage of redoing co-localization studies with newly generated vectors that encode proteins fused to eGFP/mCherry instead of mTurquoise2/mCitrine. Although the pseudo-coloring of the images can be easily changed from blue/yellow to green/red, we feel that this would cause confusion as to which

fluorophore (eGFP/mCherry or mTurquoise2/mCitrine) was used in the specific experiment. Thus, we provide the images in green/red for reviewers use.

The clear results for Pearson's correlation coefficient that seemingly differ from the co-localization images result from the fact that the shown images are maximum projections of all detected z-planes whereas the Pearson's correlation coefficient was calculated from individual (middle) sections. We have now included this information in the materials&methods section and in the figure legends.

2) While cell death induction by WT EGFP-BOK is inhibited by mCherry-BCL-2 co-expression, cell death induced by EGFP-BOK;BAX-TMD was not altered by co-expression of mCherry-BCL-2 (Figure 6B). The authors interpret this by stating that BCL-2 cannot inhibit BOK localized on mitochondria (which seems to be the case based on Figure S3C). However, two control experiments are missing. 1) In Figure S3C it is not really shown that upon co-expression with EGFP-BOK;BAX-TMD, mCherry BCL-2 remains ER-bound. The co-staining picture is not clear. 2) Since BAX recruits BCL-2 to mitochondria a BH3 and not a TMD interaction, this does not seem to be the case with EGFP-BOK;BAX-TMD. To prove this, the authors should express the a BH3-L70E mutant of EGFP-BOK;BAX-TMD to show that it also does not attract mCherry-BCL-2 to mitochondria. This mutant would also be interesting for functional assays in Figure 6 to show if EGFP-BOK-mediated cell death is indeed BH3-dependent (both on the ER and on mitochondria). If cell death is independent of BH3, then it should still be inhibited by mCherry-BCL-2 via TMD interaction.

We thank the reviewer for the insightful comments on this specific aspect. We agree that Fig. S3C does not allow to differentiate whether BCL-2 localizes to Mitochondria or the ER. In accordance with the reviewer we also feel that mCherry-BCL-2 partially localizes to mitochondria similar to eGFP-BOK;BAX-TMD. We would like to point out that the separated BCL-2-TMD localizes to the ER while (full length) mCherry-BCL-2 frequently is found at the mitochondria – probably due to interaction with endogenous Bcl-2 family proteins (as discussed in the manuscript likely via the BH3:hydrophobic groove interaction).

As suggested by the reviewer, we have now additionally analyzed the co-expression of eGFP-BOK(L70E);BAX-TMD with mCherry-BCL-2. In these experiments BH3-mutation in eGFP-BOK(L70E);BAX-TMD does not affect localization of mCherry-BCL-2. In line, cell death induced by eGFP-BOK(wt);BAX-TMD and eGFP-BOK(L70E);BAX-TMD is not affected by mCherry-BCL-2.

We have included these additional images (of eGFP-BOK(L70E);BAX-TMD) in expanded view data (EV7) and for the reviewer use (below).

3) While EGFP-BOK;BAX-TMD expression led to more Annexin-V+ apoptotic DU145 cells than EGFP-BOK expression (Figure 6B) due to a higher stability of mitochondrially localized EGFP-BOK;BAX-TMD, this difference is not seen when apoptosis was measured by caspase-3 activity (Figure 6F). Why is there a discrepancy? Does this mean that EGFP-BOK;BAX-TMD majorly induces caspase-independent cell death when targeted to mitochondria? This is an interesting finding that needs to be discussed.

Again, we thank the reviewer for his/her well-thought and justified concern. Several aspects contribute to this “discrepancy”: Fig. 6B shows results from flow cytometric analysis, shown is % of eGFP+ cells that are Annexin V+. In contrast, the caspase activity assay is not restricted to eGFP+ cells and includes non-transfected cells.

To address the reviewer's concern we analyzed whether eGFP-BOK(wt) and eGFP-BOK;BAX-TMD induced cell death of DU145 cells is caspase dependent. These experiments show that caspase-inhibition by QVD-OPh similarly blocks cell death induced by either eGFP-BOK(wt) and eGFP-BOK;BAX-TMD. We include these new data in expanded view figure EV7 (below).

Minor points:

Lines 235-236: The MCL-1-TMD data are not shown in Figure S2A. This should be indicated.

Lines 282-293: The co-localization studies are shown in Figure 4C and Figure S3A and not S4A).

We thank the reviewer for the detailed inspection and have corrected our mistakes.

Referee #2:

This manuscript provides a very detailed interrogation of the binding partners of the transmembrane domain (TMD) of the Bcl2 family member BOK and provides a structural basis for why this protein behaves differently from other members of the family. For the most part experiments are well presented and controlled with suitable quantitation. The major weakness is in Figure 6 where, as detailed below, the effects on apoptosis are just too small to provide meaningful conclusions on the role or necessity of BOK for apoptosis.

We thank the reviewer for the appreciation of our work and relevant concerns. We are confident that we satisfactorily reply to each of the raised concerns as detailed below.

Major concerns.

- 1. There are 19 supplemental figures. Some of them should be included in the main figures. For example S1B would be better than the current 1C as the images only show that the linked flours are expressed. Another example is Figure S2F. If others could be consolidated into fewer figures it may make it a little easier to follow.*

We thank the reviewer for this helpful instruction. We have adjusted the figures accordingly. We now include luminescence data of S1B in figure 1 and have moved figure 1D to the supplements. We also include S2F in the main figures. We also see the problem of extensive supplemental data and have consolidated the supplements for easier comprehension.

- 2. Most of the imaging comes with very nice quantitation except for Figure 4. It seems that the co-localization coefficients are only representing 1 image. Quantitation similar to the other figures should be included. Also, it is stated on line 281 that analysis was only done on cells without clustered EGFP signals. What proportion of cells does this represent? Similarly how were the high vs low levels of Bax determined?*

We agree with the reviewer and have added quantification as requested. The data is included as expanded view figure EV4. Concerning the proportion of cells with clustered EGFP signals after overexpression of EGFP-BAX/EGFP-BAK, we detected clusters in approx. 70 % of cells. Cells with EGFP-BAX clusters were assumed high expressing – cells with diffuse cytosolic distribution of EGFP-BAX were determined as cells with low BAX levels. This discrimination is in line with the relative intensity of the detection signal in the respective cells.

3. *I am not a structural biologist so I cannot fully critique the methods and interpretations of the molecular docking data. It would have been nice to test some mutants of the identified residues to validate the model but I agree that this may be outside the scope of the current manuscript.*

We appreciate that the reviewer is interested in these data (similar to reviewer #3). We also thought it would be nice to test some mutants and have done extensive cloning to prepare vectors for the expression of TMDs with amino acid substitutions at identified contact sites. We prepared the mutant BCL-2-TMDs (included in figure 6) and also BOK-TMDs (A192F; A203F A204F). All mutants showed significantly reduced signals in the NanoBiT assay, however, Western Blot analysis revealed diminished expression of mutant BOK-TMDs (LgBiT-BOK-TMD^{A/F} and LgBiT-BOK-TMD^{AA/FF}). We include results for subcellular localization and NanoBiT assays using mutant BCL-2-TMDs in the new main figure 6. Mutation of the BCL-2-TMD diminishes/abolishes luminescence in the split luciferase assay indicating abrogated interaction with BOK-TMD corroborating our prediction of the interaction interface.

For the reviewer's information we provide the results for mutant BOK-TMDs that were omitted from publication. Mutation of the chosen alanine residues to phenylalanine in BOK-TMD resulted in reduced and apparently abrogated expression of the reporter fusion protein. Irregardless, we feel that the mutant BCL-2-TMDs convincingly corroborate involvement of the identified amino acids in interaction of BOK-TMD and BCL-2-TMD.

A**B****C**

4. Figure 6 uses mostly overexpression to assess BOK induced apoptosis. The problem is that the amount of death induced by BOK is not very good ranging from 10-20% in DU145 cells or not very well inhibited by Bcl2 at around 10%. Although these differences are statistically significant, they are likely not biologically relevant especially when Bcl2 expression usually induces full resistance in other contexts. Figure 6C is particularly problematic because instead of showing the raw data the corrected percentage inhibition is shown. This could equally be 100% cell death going down to 5% or 5% going down to 1 which is not as effective. Due to the small effects, one could conclude that there are issues with the overexpression-based system not fully recapitulating the functions of these proteins OR that the BOK and BCL2 interaction regulates something other than cell death. It is difficult to fully conclude either based on the data as shown. Investigating the Bok dependence of cell death induced by different stimuli using CRISPR or RNAi with or without Bcl2 may be a better approach.

We thank the reviewer for this professional evaluation and specific comments. We see that apoptosis induction by overexpression of BOK (or any other pro-apoptotic protein) usually is more pronounced in BAX&BAK proficient cell systems which thereby possess a baseline ‘priming’ for apoptosis (e.g. Bhatt et al., 2020; doi: 10.1016/j.ccell.2020.10.010) and/or due to feedback mechanisms. Also, the threshold mediated by endogenously expressed anti-apoptotic proteins is higher in BAX&BAK negative cells. Frankly, in our experience overexpression of BOK in general is rather inefficient as compared to any other anti- or pro-apoptotic Bcl-2 protein. We have, however, included cell death induction by eGFP-BOK in DU145 after 42h (EV7D) showing pronounced eGFP-BOK induced apoptosis that is similarly reduced by mCherry-BCL-2. Of course, the suggested approach to analyze the effect of BOK knock-down/out on the response to various cell death stimuli would be elegant and add to the physiologic relevance of BOK – but only indirectly add to the topic of the present manuscript, i.e. interaction of Bcl-2 transmembrane domains in general and more importantly those of BOK and BCL-2 specifically. Especially mutation of the TMDs and their effect on apoptosis induction is more efficiently investigated in overexpression experiments than in CRISPR/Cas9 mediated knock-in approaches. We hope that the reviewer can agree with our line of thought and that these additional experiments are beyond the scope of the underlying manuscript.

Minor

1. A number of the supplemental figures are incorrectly labeled in the text

We thank the reviewer for detailed reading of our manuscript and we double-checked and corrected figure references.

2. In Figure 2D, it would be helpful to include a gate on what is considered double positive cells in the flow plots

We thank the reviewer for this helpful recommendation and adjusted the flow plots accordingly.

3. Some of the confocal images appear a little oversaturated especially the EGFP and TFP images. Suggest dialing them back a little

We thank the reviewer for the suggestion and adjusted brightness/contrast of respective images.

4. What is the rationale for using the Bax/Bak DKO cells?

We thank the reviewer for this relevant question and are glad to elaborate on our intention. It is likely that TMDs of endogenous BAX/BAK are able to interact with exogenously expressed TMD peptides. At least BAX homotypic TMD interaction is a well-studied feature of BAX pore formation (PMID 26702098) which was the rationale to use BAX TMD homotypic interaction as a positive control in our interaction assay. Therefore, we sought to exclude that endogenous BAX/BAK expression influences TMD localization. Confirming this, TMD localization in BMK/DKO cells is consistent with localization found in BAX/BAK-proficient MCF-7 cells.

5. The intro is a little too long and the whole manuscript could use a good proofread, otherwise it is nicely written

We highly appreciate that the reviewer deems our manuscript “nicely written”. We gladly followed the reviewer’s advice to shorten the introduction trying not to compromise comprehensibility of the manuscript. We explicitly demanded rigorous proofreading by each co-author to make sure all text mistakes are removed.

Referee #3:

Since Bcl-2 was discovered as an oncogene inhibiting cancer cell apoptosis in the 80s, and cytochrome c was determined as an activator of caspases and was released from mitochondria by Bax in the 90s, mitochondrial dependent apoptosis regulated by Bcl-2 and other family proteins has been the focus for decades. As a result, the molecular mechanism by which Bcl-2s act to control the protein permeability of the mitochondrial membrane has been largely established. Structure and function-based translational research has resulted in a small molecule Bcl-2 inhibitor that has been used to treat blood cancer patients. In contrast, although Bcl-2s were known early on not only localizing to the mitochondria but other intracellular membranes, research of Bcl-2s structure and function in the membranes other than the mitochondrial ones has not been proportional. In particular the interactions within the Bcl-2 family were poorly defined, and the consequences of the interactions were mostly unknown.

In the manuscript submitted to EMBO Report, Beigl et al. concluded that Bcl-2 and Bok regulate apoptosis by interaction of their C-terminal transmembrane domains (TMDs) embedded in the ER membrane. The conclusion was based on the following evidence.

- 1. The interaction between the TMDs (not as parts of the full-length proteins) occurred in cells as revealed by a novel biomolecular split luciferase assay.*
- 2. The TMDs were colocalized at the ER in cells as imaged by super-resolution fluorescence microscopy.*
- 3. The TMDs in the full-length proteins were critical for their colocalization at the ER because the colocalization was reduced when the TMD of Bcl-2 was replaced by the TMD of TOM5 that localized Bcl-2 to the mitochondria.*
- 4. The BH3 domain of Bok and the BH3-binding groove of Bcl-2 was not critical to the colocalization because if this binding interaction still occurred between the Bcl-2 mutant with the TOM5 TMD, the Bok at the ER would attract the Bcl-2 mutant to the ER, or vice versa. A corroborating evidence is that mutation of a conserved L70 residue in the BH3 domain to E that would reduce the BH3-groove interaction did not affect the colocalization Bok and Bcl-2, and that deletion of the TMD of Bok resulted in a cytosolic localization of Bok in the presence of Bcl-2 that are in the ER and mitochondria.*
- 5. Structural models for BOK-Bcl-2 TMD interactions in an ER mimic lipid bilayer were generated by molecular dynamics simulations. These models suggest that the TMDs can not only form hetero dimers and oligomers but homo dimers and oligomers.*
- 6. Distribution of certain ER lipids in the proximity of the TMD dimers and oligomers is different than the bulk lipid bilayer suggesting their preferential interactions with the TMD complexes.*
- 7. The TMDs and their colocalization to the ER are important to Bcl-2 inhibition of Bok-induced apoptosis as determined by combinations of Bcl-2 and Bok mutants with different TMDs.*

Thus, the overall conclusion is mostly based on experimental evidence. Moreover, molecular dynamics simulations provide structural models of molecular to atomic resolutions for the TMD complexes in the ER lipid bilayer.

Therefore, the manuscript would be suitable for publication. However, the following concern needs to be addressed to further solidify the conclusion and increase the impact of the work.

The authors are very grateful for the sophisticated evaluation by this reviewer. We thank the reviewer for the positive evaluation of our work and are confident that we satisfactorily address the concerns in the following point-to-point reply.

As part of the overall conclusion, the interaction of Bcl-2 and Bok that is mediated by their TMDs in the ER membrane is critical to the Bcl-2 inhibition of Bok-induced apoptosis. The work is then one step short from a full demonstration of the critical role of "the interaction between the TMDs in the ER membrane". What the authors might do is using the structural models of the Bcl-2-Bok TMD complexes generated by molecular dynamics simulations to identify point mutations in the complex interfaces, and then determine if the mutations reduce the complex formation in the ER membrane and the Bcl-2 inhibition of Bok in cells.

We appreciate that the reviewer is interested in these data (similar to reviewer #2). In order to step further into a full demonstration of the interaction between the BCL-2-TMD and BOK-TMD we followed the reviewer's advice and prepared vectors for the expression of TMDs with amino acid substitutions at identified contact sites. We prepared the mutant BCL-2-TMDs (included in figure 6) and also BOK-TMDs (A192F; A203F A204F).

All mutants showed significantly reduced signals in the NanoBiT assay, however, Western Blot analysis revealed diminished expression of mutant BOK-TMDs (LgBiT-BOK-TMD^{A/F} and LgBiT-BOK-TMD^{AA/FF}). We include results for subcellular localization and NanoBiT assays using mutant BCL-2-TMDs in the new main figure 6. Mutation of the BCL-2-TMD diminishes/abolishes luminescence in the split luciferase assay indicating abrogated interaction with BOK-TMD corroborating our prediction of the interaction interface.

For the reviewer's information we provide the results for mutant BOK-TMDs that were omitted from publication. Mutation of the chosen alanine residues to phenylalanine in BOK-TMD resulted in reduced and apparently abrogated expression of the reporter fusion protein. Irregardless, we feel that the mutant BCL-2-TMDs convincingly corroborate involvement of the identified amino acids in interaction of BOK-TMD and BCL-2-TMD.

In addition, the following questions need to be answered during revision of the manuscript.

- A. *Is there any structural model for how Bok and Bcl-2 interacts via the BH3-groove interface? If so, would the L70E mutation disrupt the complex? If not, could a such model be generate using AlphaFold2 or other AI structural biology tools? If still not, perhaps adding ABT-199 to the cells where Bcl-2 can inhibit Bok is a better experiment to do to see if the BH3-groove interaction mediates the Bok/Bcl-2 interaction and function.*

These are highly sophisticated questions which we sadly can not completely address.

1st: we are not aware of a model for BOK:BCL-2 interaction via BH3:hydrophobic groove interaction. In fact, our data indicates that there is no such interaction.

2nd. It is assumed that the L70E mutation abrogates the potential BH3:hydrophobic groove interaction. Since we do not see an interaction, we also do not see an influence of the L70E mutation. Frankly, the causality is presented reversely in our manuscript – since we do not see an influence of the L70E mutation we assume that there is no interaction.

3rd. We are sure such a model could be generated but we feel that the data from wet lab experiments in our manuscript are superior to computer generated “results”. No offense.

4th. We happily followed the suggestion to analyze the influence of ABT-199 on the co-localization of eGFP-BOK and mCherry-BCL-2 (image for reviewer’s use). The images show that ABT-199 does not affect co-localization of eGFP-BOK and mCherry-BCL-2 – corroborating the assumption that the interaction is not mediated by BOK BH3 binding to the hydrophobic groove of BCL-2. We also included data for BOK(L70E);BAX-TMD (requested by reviewer #1) in the manuscript (Fig. EV7). Also here the L70E mutation had no effect.

B. *In the experiments for Figure 2, was any Bcl-2-TMD localize to the mitochondria? If not, please explain the difference between the TMD in the full-length Bcl-2 protein and the TMD by itself on the subcellular localization.*

We have some problems to interpret the reviewer's question. Some of the TMDs from Bcl-2 proteins localized to mitochondria (BCL-xL, BAX, BAK). The TMD of the BCL-2 protein (predominantly) localized to the ER. However, we cannot rule out that some of the BCL-2-TMD is also localized to mitochondria. We have included a comprehensive table of results from our localization studies for the reviewer's use and hope to convince the reviewer of the validity of our data. We and others have experienced that the interaction of Bcl-2 proteins greatly affects the localization of the full length protein (e.g. PMID: 36603764).

Sub-type	Protein TMD	Predominant localisation (TMD peptide)			Localisation of full-length protein (normal conditions)
		 Mito	 ER	 Others	
Effector	BAX				Cytosol
	BAK				Mitochondria
	BOK				ER
Anti-apoptotic Bcl-2-like	BCL-2				ER
	BCL-XL				Mito/Cytosol
	BCL-W				Cytosol
	MCL-1				Mitochondria
	A1			Golgi	Mitochondria

C. *There are many colocalizations compared throughout the text in Results using correlation coefficients. It would be better if they can be tabalized.*

We thank the reviewer for his helpful suggestion. We include tables of summarized correlation coefficients in supplements (Appendix Table S5/6).

D. *What would be the effect of endogenous Bok and Bcl-2 on the experiments performed with different cells?*

We thank the reviewer for the insightful comment. We do not expect that the endogenously expressed proteins significantly affect the results obtained.

We specifically chose DU145 for the main part of our experiments since these cells do not express BAX and also not BCL-2. Expression of BAK is low compared to other cell lines. We sought to address the effect of endogenously expressed “TMDs” on the NanoBiT interaction assay (interception of LgBiT-TMD or SmBiT-TMD) by using different cell lines (MCF7, HEK293). Results generated in MCF7 and HEK293 are similar likely indicating negligible influence of endogenously expressed BOK or BCL-2. Although interaction with endogenous proteins cannot be ruled out completely, we feel that a dominant role of endogenous “TMDs” is unlikely. We gladly offer to address this point with regard to figure EV7A in more detail in the discussion part of the manuscript.

E. *Line 271-273, add another reference for Bak/Bcl-2 interaction via the BH3/groove binding interface.*

We thank the reviewer for his detailed revision of the manuscript. We now include Willis *et al.* (2015), PMID: 15901672 in the manuscript.

F. *Some of the ER lipids would be enriched or depleted around the Bcl-2/Bok TMD complexes according to molecular dynamics simulations. Could this prediction be tested by experiments?*

This is a very interesting question that we would like to answer in future experiments. However, at this point we feel that this is beyond the scope of the manuscript and, frankly, currently also our expertise. We would assume that a possible approach is immunoprecipitation of tagged TMDs after crosslinking and subsequent mass-spectrometric analysis. We would happily include an outlook regarding this aspect in the discussion part of the manuscript.

G. *Adding a schematic overall model for how Bcl-2 interacts with Bok at the ER to regulate Bok-induced apoptosis would be better for enhancing the visibility of this new discovery.*

We very much appreciate this suggestion of the reviewer and now include a schematic as a synopsis to our manuscript.

H. *The following papers could be referenced in the manuscript as early computational modeling and molecular dynamics simulations of Bax TMD homodimers in the mitochondrial lipid bilayer.*

1) Zhang Z, Subramaniam S, Kale J, Liao C, Huang B, Brahmabhatt H, Condon SG, Lapolla SM, Hays FA, Ding J, He F, Zhang XC, Li J, Senes A, Andrews DW, Lin J. BH3-in-groove dimerization initiates and helix 9 dimerization expands Bax pore assembly in membranes.

EMBO J. 2016 Jan 18;35(2):208-36. doi: 10.15252/embj.201591552. Epub 2015 Dec 23. PMID: 26702098; PMCID: PMC4718459.

2) Liao C, Zhang Z, Kale J, Andrews DW, Lin J, Li J. Conformational Heterogeneity of Bax Helix 9 Dimer for Apoptotic Pore Formation. *Sci Rep.* 2016 Jul 6;6:29502. doi: 10.1038/srep29502. PMID: 27381287; PMCID: PMC4933972.

We thank the reviewer for the comment and appreciate the profound knowledge of published data. We included the respective references appropriately.

EMBOR-2023-58132-T

BCL-2 and BOK regulate apoptosis by interaction of their C-terminal transmembrane domains

Additional data for reviewer #1

- 1. Images in red/green pseudo color**
- 2. Additional images for eGFP-BOK;BAX-TMD + mCherry-BCL-2**

1.

Fig 2A

Fig 2B

Scale bar = 10 μ m.

1.

Fig EV2 (S2)

Fig EV2 (S2)

Scale bar = 10 μ m.

1.

Fig 3

Scale bar = 10 μ m.

1.

Fig EV4 C (S3A)

Scale bar = 10 μ m.

1.

Fig 6

Scale bar = 10 μ m.

2.

Additional images figure EV7G (S3C)

MCF-7 cells were transfected with plasmids encoding for EGFP-BOK^{BAX-TMD} and mCherry-BCL-2. After 24 h, cells were fixed and mounted with DAPI-containing mounting medium followed by cLSM. Images are maximum projections of z-stacks. Representative images from n = 2 independent experiments. Scale bar = 10 μ m.

EMBOR-2023-58132-T

BCL-2 and BOK regulate apoptosis by interaction of their C-terminal transmembrane domains

Additional data for reviewer #2

- 1. Exemplary cell with/without BAX clusters**
- 2. Interaction data + Western Blot of BOK-TMD and BCL-2-TMD mutants**

1.

BAX clusters

diffuse BAX

Representative cell (MCF-7) expressing EGFP-BAX. Color scale reflects differences in EGFP fluorescence intensity. Maximum projections of z-stacks. Scale bar = 10 μm .

2.

Split-luciferase assay in HEK293FT cells transfected with plasmids for the expression of indicated NanoBiT-TMDs. Cells were harvested after 24 h and samples were both used for split-luciferase assay and Western Blot. BAX-TMD/TOM5-TMD serves as a negative control, while BAX-TMD/BAX-TMD serves as positive control. Mutants of SmBiT-BCL-2-TMD were combined with LgBiT-BOK-TMD (WT = wild-type). Graphs show luminescence intensity relative to the positive control and normalized to mTurquoise2 fluorescence intensity. Shown is the mean \pm sd from three or four independent experiments. In Western Blots, corresponding whole cell lysates were analyzed for expression of LgBiT and mTurquoise2 (mTurq2). Representative blot from two independent experiments is shown.

EMBOR-2023-58132-T

BCL-2 and BOK regulate apoptosis by interaction of their C-terminal transmembrane domains

Additional data for reviewer #3

- 1. Additional images for EGFP-BOK + mCherry-BCL-2 upon ABT-199 incubation**
- 2. Interaction data + Western Blot of BOK-TMD and BCL-2-TMD mutants**
- 3. Table of TMD subcellular localization**

1.

MCF-7 cells were transfected with plasmids encoding for EGFP-BOK and mCherry-BCL-2 and incubated with 5 μ M ABT-199. After 24 h, cells were fixed and mounted with DAPI-containing mounting medium followed by cLSM. Images are maximum projections of z-stacks. Representative images from n = 2 independent experiments. Scale bar = 10 μ m.

2.

Split-luciferase assay in HEK293FT cells transfected with plasmids for the expression of indicated NanoBiT-TMDs. Cells were harvested after 24 h and samples were both used for split-luciferase assay and Western Blot. BAX-TMD/TOM5-TMD serves as a negative control, while BAX-TMD/BAX-TMD serves as positive control. Mutants of SmBiT-BCL-2-TMD were combined with LgBiT-BOK-TMD (WT = wild-type). Graphs show luminescence intensity relative to the positive control and normalized to mTurquoise2 fluorescence intensity. Shown is the mean \pm sd from three or four independent experiments. In Western Blots, corresponding whole cell lysates were analyzed for expression of LgBiT and mTurquoise2 (mTurq2). Representative blot from two independent experiments is shown.

3.

Sub-type	Protein TMD	Predominant localisation (TMD peptide)			Localisation of full-length protein (normal conditions)
		 Mito	 ER	 Others	
Effector	BAX				Cytosol
	BAK				Mitochondria
	BOK				ER
Anti-apoptotic Bcl-2-like	BCL-2				ER
	BCL-XL				Mito/Cytosol
	BCL-W				Cytosol
	MCL-1				Mitochondria
	A1			Golgi	Mitochondria

Summary of TMD peptide subcellular localization in comparison to subcellular localization of full length proteins.

Dear Dr. Essmann

Thank you for the submission of your revised manuscript to EMBO reports. We have now received the full set of referee reports that is copied below.

As you will see, all referees acknowledge that the manuscript has been significantly strengthened during the revision. However, referee #2 and #3 raise some remaining concerns. These concerns relate to the functional evidence provided that the interaction via the TMDs is relevant to apoptosis induction and the interaction in the ER. Please address these remaining concerns by discussing the limitations of the experiments and by reanalysing the data shown in Figure 6C and 7C. Please also determine the contribution of the TMD interface to apoptosis by testing whether the BCL-2 TMD mutant versions are able to inhibit BOK, as suggested by referee #3. These experiments seem feasible and would considerably strengthen the functional relevance of the TMD interaction.

From the editorial side, there are also a few things that we need before we can proceed with the official acceptance of your study.

- Please provide up to 5 keywords.
- Please rename the Conflict of Interest statement to "Disclosure Statement and Competing Interests" and place it after the Acknowledgments section.
- EV figure legends need to be bundled and placed after the main figure legends in a consecutive order. The heading is "Expanded View Figure Legends".
- The EV Figures are not properly numbered/labeled/called out: EV1, EV2, EV4, EV6, EV7. Please correct the numbering.
- Please note the following name discrepancy: Markus Rehm in the manuscript file versus Markus Morrison (Rehm) in the online manuscript tracking system. Please correct either of them.
- Please add the following funding information in the manuscript tracking system: Stuttgart Center for Simulation Science (SC SimTech). The information in the system will be used for publication.
- Appendix: Please add page numbers to the table of content.
- Please number Appendix Table S1A and S1B as Appendix Table S1 and S2, respectively, and change the numbering of the subsequent tables accordingly.
- Please note that all materials and methods must be part of the main manuscript text.
- Please upload the source data as one folder per figure.
- Our production/data editors have asked you to clarify several points in the figure legends (see below). Please incorporate these changes in the manuscript and return the revised file with tracked changes with your final manuscript submission.
 - A. Please note that a separate 'Data Information' section is required in the legends of figures 7a-c, e-h.
 - B. Please note that the legends for figures EV 4c-g is not provided in the sequential manner (legend for figures EV 4d-e, g-h are provided before legend of figures EV 4c, f). This needs to be rectified.
 - C. Please note that the legends for figures EV 5 and EV 6 are labelled as EV 6 and EV 7 respectively. This needs to be rectified.
 - D. Please define the annotated p values ****/***/**/* in the legend of figure 1f; 6c; 7a-c, e-h; EV 1e; EV 7c-d; as appropriate.
 - E. Please indicate the statistical test used for data analysis in the legends of figures 1f; 6c; 7a-c, e-h; EV 1e; EV 7c-d.
- As a standard procedure, we edit the title and abstract of manuscripts to make them more accessible to a general readership. Please find the edited suggestion below my signature.
- EMBO Reports papers are accompanied online by A) a short (1-2 sentences) summary of the findings and their significance, B) 2-3 bullet points highlighting key results. Please send us this information along with the revised manuscript.
- The final size of the synopsis image will be small (550 pixels width). At that size, the text is rather small and cannot be read easily. Please check the image at this size and enlarge the text accordingly.
- On a different note, I would like to alert you that EMBO Press offers a new format for a video-synopsis of work published with us, which essentially is a short, author-generated film explaining the core findings in hand drawings, and, as we believe, can be

very useful to increase visibility of the work. This has proven to offer a nice opportunity for exposure i.p. for the first author(s) of the study. Please see the following link for representative examples and their integration into the article web page:

<https://www.embopress.org/doi/full/10.15252/emboj.2019103932>

With kind regards,

The Bcl-2 family controls apoptosis by direct interactions of pro- and anti-apoptotic proteins. The BH3 domain of pro-apoptotic Bcl-2 family proteins binds to the hydrophobic groove of their anti-apoptotic siblings, liberating apoptosis effector proteins and inducing cell death, which is therapeuti. Evidence suggests that also the transmembrane domain (TMD) of Bcl-2 proteins affects Bcl-2 interactions. We developed a highly-specific split luciferase assay enabling the analysis of TMD interactions of pore-forming apoptosis effectors BAX, BAK, and BOK with anti-apoptotic Bcl-2 proteins in living cells. We confirm homotypic interaction of the BAX-TMD, but also newly identify interaction of the TMD of anti-apoptotic BCL-2 with the TMD of BOK, a peculiar pro-apoptotic Bcl-2 protein. BOK-TMD and BCL-2-TMD interact at the endoplasmic reticulum. Molecular dynamics simulations support dynamic BOK-TMD and BCL-2-TMD dimers and stable heterotetramers. Mutation of BCL-2-TMD at predicted key-residues abolishes interaction with BOK-TMD. Inhibition of BOK-induced apoptosis by BCL-2 depends specifically on their TMDs, as a chimeric BOK protein with the BAX TMD is insensitive to inhibition by BCL-2. Thus, our data reveal that the TMDs of Bcl-2 proteins are a relevant interaction interface for apoptosis regulation.

Referee #1:

The authors have satisfactorily responded to my criticisms. The MS can now be accepted

Referee #2:

In the revised manuscript by Beigl et al the authors addressed most of my critiques except what I viewed as the major weaknesses of the paper - the functional data showing that BOK-induced apoptosis is inhibited by BCL2 via the TMD interactions. They argued that the experiments I suggested are beyond the scope of the manuscript and would only indirectly add to the topic of the manuscript. However, the experiments showing a functional effect of the BOX-BCL2 interaction are essential to showing that the TMD interaction has a relevance and the data shown shows minor effects at best. At the very least Figure 7C should be removed as, in my opinion, the corrected values shown are difficult to interpret. Secondly, since the authors have already laid out that the role of BOK in apoptosis is controversial, they should fully discuss the limitations of these experiments and that they don't really rule in or out a physiological role for the BOK/BCL2 TMD interaction in apoptosis.

Referee #3:

I thank that the authors have made an effort to address my requests and questions. I am satisfied with their answers with only one exception regarding the following request.

As part of the overall conclusion, the interaction of Bcl-2 and Bok that is mediated by their TMDs in the ER membrane is critical to the Bcl-2 inhibition of Bok-induced apoptosis. The work is then one step short from a full demonstration of the critical role of "the interaction between the TMDs in the ER membrane". What the authors might do is using the structural models of the Bcl-2-Bok TMD complexes generated by molecular dynamics simulations to identify point mutations in the complex interfaces, and then determine if the mutations reduce the complex formation in the ER membrane and the Bcl-2 inhibition of Bok in cells. To address this request, the authors generated Bcl-2 and Bok TMD mutants to address to validate the interface between the

TMDs indicated by MD simulations. Based on the Western blot data shown in the new Figure 6C, the LgBiT-Bok TMD was expressed at different levels depending on which SmBiT-Bcl-2 TMD was coexpressed in the cells. For example, the LgBiT-Bok TMD was expressed at a similar level when it is coexpressed with the wild type or LVI/AAA mutant SmBiT-Bcl-2 TMD. In contrast, the LgBiT-Bok TMD was expressed at a lower level when it is coexpressed with the other three mutant SmBiT-Bcl-2 TMDs. Since the interaction between the Bcl-2 and Bok TMDs would be dependent on their expression levels in the cells or concentrations in the ER membranes, the authors' conclusion about the LVI/AAA mutation is correct whereas the conclusion about the other three mutations may be not.

Moreover, the authors did not determine if the mutations in the Bcl-2 TMD that reduced the interaction with the Bok TMD in the new Figure 6C reduce the Bcl-2 inhibition of Bok in cells when the same mutations are put into the TMD of full-length Bcl-2.

Referee #1:

The authors have satisfactorily responded to my criticisms. The MS can now be accepted.

We are grateful for this reviewer's kind and positive evaluation!

Referee #2:

In the revised manuscript by Beigl et al the authors addressed most of my critiques except what I viewed as the major weaknesses of the paper - the functional data showing that BOK-induced apoptosis is inhibited by BCL2 via the TMD interactions. They argued that the experiments I suggested are beyond the scope of the manuscript and would only indirectly add to the topic of the manuscript. However, the experiments showing a functional effect of the BOK-BCL2 interaction are essential to showing that the TMD interaction has a relevance and the data shown shows minor effects at best. At the very least Figure 7C should be removed as, in my opinion, the corrected values shown are difficult to interpret. Secondly, since the authors have already laid out that the role of BOK in apoptosis is controversial, they should fully discuss the limitations of these experiments and that they don't really rule in or out a physiological role for the BOK/BCL2 TMD interaction in apoptosis.

We thank the reviewer for his professional criticism. We have addressed each of these points as detailed below:

At the very least Figure 7C should be removed....

We have removed Figure 7C to avoid misleading presentation and over-interpretation of data.

Secondly, since the authors have already laid out that the role of BOK in apoptosis is controversial, they should fully discuss the limitations of these experiments and ...

The reviewer is correct, we explicitly mention that the exact function of BOK in apoptosis is controversial. Instead of extensively discussing the principle role of BOK in apoptosis, we now have explicitly mentioned the limitation of the overexpression experiments. We make clear, that overexpression of BOK was less efficient than overexpression of other Bcl-2 proteins. In this context we point out, that data shown is explicitly limited to cells that express EGFP-BOK at a detectable level (lines 475 – 477).

that they don't really rule in or out a physiological role for the BOK/BCL2 TMD interaction in apoptosis.

We appreciate the suggestion to extent the discussion part of the manuscript. The (newly) included data show that the TMD-interaction clearly affects apoptosis induction in overexpression experiments. Since data stem from overexpression experiments, we discuss that a role in physiological context is potential and present hypotheses how interaction of BOK-TMD and BCL-2-TMD could fine-tune apoptosis regulation.

Referee #3:

I thank that the authors have made an effort to address my requests and questions. I am satisfied with their answers with only one exception regarding the following request.

As part of the overall conclusion, the interaction of Bcl-2 and Bok that is mediated by their TMDs in the ER membrane is critical to the Bcl-2 inhibition of Bok-induced apoptosis. The work is then one step short from a full demonstration of the critical role of "the interaction between the TMDs in the ER membrane". What the authors might do is using the structural models of the Bcl-2-Bok TMD complexes generated by molecular dynamics simulations to identify point mutations in the complex interfaces, and then determine if the mutations reduce the complex formation in the ER membrane and the Bcl-2 inhibition of Bok in cells.

To address this request, the authors generated Bcl-2 and Bok TMD mutants to address to validate the interface between the TMDs indicated by MD simulations. Based on the Western blot data shown in the new Figure 6C, the LgBiT-Bok TMD was expressed at different levels depending on which SmBiT-Bcl-2 TMD was coexpressed in the cells. For example, the LgBiT-Bok TMD was expressed at a similar level when it is coexpressed with the wild type or LVI/AAA mutant SmBiT-Bcl-2 TMD. In contrast, the LgBiT-Bok TMD was expressed at a lower level when it is coexpressed with the other three mutant SmBiT-Bcl-2 TMDs. Since the interaction between the Bcl-2 and Bok TMDs would be dependent on their expression levels in the cells or concentrations in the ER membranes, the authors' conclusion about the LVI/AAA mutation is correct whereas the conclusion about the other three mutations may be not.

Moreover, the authors did not determine if the mutations in the Bcl-2 TMD that reduced the interaction with the Bok TMD in the new Figure 6C reduce the Bcl-2 inhibition of Bok in cells when the same mutations are put into the TMD of full-length Bcl-2.

We thank the reviewer for his detailed and professional revision of our manuscript and data. We very much appreciate his overall positive evaluation. We have generated additional functional data to substantiate the direct interaction of BOK-TMD and BCL-2-TMD and its relevance for apoptosis induction. Figure 6C shows that mutations in the BCL-2-TMD reduce luciferase activity. We now include Western Blot analysis of aliquots from cell lysates that were investigated in luciferase assays rather than independent replicate samples and show comparable expression level of each mutant TMD-reporter subunit.

As suggested by the reviewer, we now include Figure 6D which shows that overexpression of EGFP-BOK induces apoptosis and co-expression of mCherry-BCL-2 reduces the proportion of apoptotic cells (similar to Figure 7). More importantly, Figure 6D also shows that co-expression of BCL-2 with mutant TMDs less effectively reduces apoptosis induction by EGFP-BOK. These data corroborate the interaction of TMDs via the identified residues and the functional relevance of the interaction interface in apoptosis regulation – as requested by this reviewer.

Frank Essmann
Robert Bosch Center for Tumor Diseases
Apoptosis Regulation & Targeted Therapy
Auerbachstr. 112
Stuttgart, Baden-Wuerttemberg 70376
Germany

Dear Dr. Essmann,

I am very pleased to accept your manuscript for publication in the next available issue of EMBO reports. Thank you for your contribution to our journal.

Yours sincerely,
